# Neural Image Compression:
# Generalization, Robustness, and Spectral Biases

**Kelsey Lieberman**
Department of Computer Science
Duke University
Durham, NC USA
`kelsey.lieberman@duke.edu`

**James Diffenderfer**
Lawrence Livermore National Laboratory
Livermore, CA USA
`diffenderfer2@llnl.gov`

**Charles Godfrey**[†]
Thomson Reuters Labs
Eagan, MN USA
`charles.godfrey@thomsonreuters.com`

**Bhavya Kailkhura**
Lawrence Livermore National Laboratory
Livermore, CA USA
`kailkhura1@llnl.gov`

## Abstract

Recent advances in neural image compression (NIC) have produced models that are starting to outperform classic codecs. While this has led to growing excitement about using NIC in real-world applications, the successful adoption of any machine learning system in the wild requires it to generalize (and be robust) to unseen distribution shifts at deployment. Unfortunately, current research lacks comprehensive datasets and informative tools to evaluate and understand NIC performance in real-world settings. To bridge this crucial gap, first, this paper presents a comprehensive benchmark suite to evaluate the out-of-distribution (OOD) performance of image compression methods. Specifically, we provide CLIC-C and Kodak-C by introducing 15 corruptions to the popular CLIC and Kodak benchmarks. Next, we propose spectrally-inspired inspection tools to gain deeper insight into errors introduced by image compression methods as well as their OOD performance. We then carry out a detailed performance comparison of several classic codecs and NIC variants, revealing intriguing findings that challenge our current understanding of the strengths and limitations of NIC. Finally, we corroborate our empirical findings with theoretical analysis, providing an in-depth view of the OOD performance of NIC and its dependence on the spectral properties of the data. Our benchmarks, spectral inspection tools, and findings provide a crucial bridge to the real-world adoption of NIC. We hope that our work will propel future efforts in designing robust and generalizable NIC methods. Code and data will be made available at https://github.com/klieberman/ood_nic.

## 1 Introduction

Consider the Mars Exploration Rover, whose scientific objective is to search for clues to past activity of water (and perhaps life) on Mars. To achieve this, the rover collects images of interesting rocks and soils to be analyzed by the scientists on Earth. Sending these images down the Earth-bound data stream in their original form is too slow and expensive due to limited bandwidth. Thus, it is well accepted that image compression could play a key role in producing scientific breakthroughs [44].

---

[†]Work done at Pacific Northwest National Laboratory.

37th Conference on Neural Information Processing Systems (NeurIPS 2023).

Employing image compression in such a setting is challenging for three main reasons: 1) a *high compression ratio* is desired due to low communication bandwidth, 2) given the battery-operated nature of these devices, the compression module has to be *lightweight* so it consumes less memory and power, and 3) *robustness and generalization* to environmental noises and domain shifts, respectively, are desired due to limited Mars-specific training data. These requirements are not specific to the planetary exploration use case but also arise in a wide range of scientific applications which use image compression in the wild [33].

Recently, neural image compression (NIC) has demonstrated remarkable performance in terms of rate-distortion and runtime overhead on in-distribution (IND) data [9, 46]—satisfying requirements 1) and 2). However, there is limited work on understanding the out-of-distribution (OOD) robustness and generalization performance of image compression methods (requirement 3) [42]. Our work is driven by several open fundamental empirical and theoretical questions around this crucial issue.

> *How can the expected OOD performance of image compression models be reliably assessed? Can we gain a deeper understanding of the modus operandi of different image compression methods? How do training data properties and biases impact data-driven compression methods?*

**Main Contributions:** This paper takes a critical view of the state of image compression and makes several contributions toward answering the aforementioned questions. ❶ First, we design *comprehensive benchmark datasets* for evaluating the OOD performance of image compression methods. Inspired by existing OOD benchmarks for classification and detection [26, 28, 57, 56], we design CLIC-C and Kodak-C by introducing 15 common shifts emulating train-deployment distribution mismatch to the popular CLIC and Kodak datasets. ❷ Next, we focus on understanding the image compression performance. The de-facto approach is to use rate-distortion (RD) curves measured with perceptual quality metrics, such as PSNR. Such scalar metrics, although easy to compute, are known to be extremely limited in what they can capture and sometimes can even be misleading [63, 61]. To complement RD curves, we propose *spectrally-inspired inspection tools* that provide a more nuanced picture of (a) compression error, and (b) OOD performance of a given method. Specifically, we introduce a power spectral density (PSD) based approach to understand the reconstruction error. Our approach not only quantifies how much error was made but also highlights precisely where it was made (in the frequency domain). Similarly, to understand the OOD performance of a compression method in unseen deployment scenarios, we propose *Fourier error heatmaps*—a visualization tool for highlighting the sensitivity of the reconstruction performance of a compression method to different perturbations in the frequency domain. ❸ Using our benchmark datasets and inspection tools, we carry out *a systematic empirical comparison* of classic codecs (i.e., JPEG2000, JPEG, and VTM) with various NIC models (fixed rate, variable rate, and pruned versions of the scale hyperprior model [9], as well as efficient learned image compression (ELIC) [24]). ❹ Finally, we develop *theoretical tools* to connect NIC OOD performance with its training data properties.

**Main Findings:** Our analysis resulted in some invaluable insights about the state of image compression. We summarize some of our findings below.

- Compression methods yielding the same PSNR (or bpp) can produce very different spectral artifacts. Our tools help uncover hidden spectral biases and highlight the limitations of de-facto RD curve-based performance comparison.
- As the compression rate increases, different codecs prioritize different parts of the frequency spectrum. By precisely characterizing this behavior, our tools help in advancing the current understanding of the modus operandi of image compression methods.
- Image compression models generalize to low- and mid-frequency shifts better than high-frequency shifts. This finding calibrates our expectations for image compression performance on OOD data.
- NIC models are better at denoising high-frequency corruptions than classic codecs. This finding reveals that unlike classic codecs, NIC models have an inherent spectral bias due to the image frequencies present in their training data.
- Identifying the most suitable compression method becomes exceptionally challenging without the knowledge of spectral characteristics of OOD shifts. Our systematic evaluation identifies this open issue with current compression methods and suggests the design of next-generation NIC models that can adapt themselves at runtime based on the spectral nature of the data to be a potentially worthwhile direction to pursue in the future.

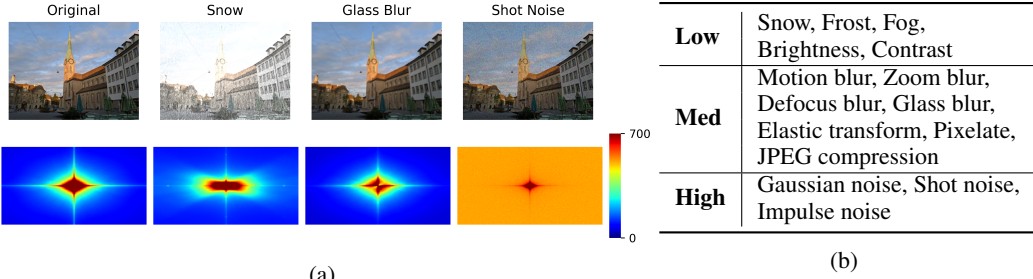

| | |
|---|---|
| **Low** | Snow, Frost, Fog, Brightness, Contrast |
| **Med** | Motion blur, Zoom blur, Defocus blur, Glass blur, Elastic transform, Pixelate, JPEG compression |
| **High** | Gaussian noise, Shot noise, Impulse noise |

(a)

(b)

Figure 1: **(a) Top row:** An original CLIC image and the same image with 3 different corruptions in CLIC-C (severity 5). **Bottom left:** Average PSD of CLIC dataset, $\frac{1}{N}\sum_{k=1}^{N} PSD(X_k)$. **Bottom row, other figures:** Average PSD of the difference between the corrupted images and the clean images for each given CLIC-C corruption $c$, $\frac{1}{N}\sum_{k=1}^{N} PSD(c(X_k) - X_k)$. **(b)** CLIC-C corruptions categorized as low, medium, or high based on corruption average $PSD$.

We corroborate our findings with a detailed theoretical analysis, showing that multiple overarching trends in our experimental results can be attributed to neural compression models' spectral bias.

*Appendix A has a detailed related work discussion, while all references are listed in the main paper.*

## 2 Out-of-distribution image compression datasets

To evaluate NIC in the presence of environmental or digital distribution shifts, we generated variants of the CLIC and Kodak datasets, which we refer to as CLIC-C and Kodak-C. Following the techniques presented in [26] for studying the performance of DNN classifiers encountering distributional shifts "in the wild", our -C datasets consist of images augmented by 15 common corruptions. For each image in the original dataset, the -C dataset contains a corrupted version of the image for each of the 15 common corruptions[2], and for each of five corruption severity levels, with 1 being the lowest severity and 5 being the highest. A sample of some corruptions on CLIC-C is provided in Figure 1a.

While each -C dataset offers a broad sampling of environmental or digital image corruptions, it also provides a spectrally diverse collection of corruptions, in the sense that each corruption can be categorized as low, medium, or high frequency based on the frequency content used for perturbations. We will write $PSD(\cdot)$ to denote the function that converts the input image from the spatial to the frequency domain by computing the power spectral density of the input. Practically, computing $PSD(\cdot)$ is done by applying the fast Fourier transform (FFT) [11], followed by a shift operation to center the zero-frequency component, then taking the absolute value. Now suppose we have a set $\mathcal{X} = \{X_k\}_{k=1}^{N}$ of uncorrupted images and some corruption function $c(\cdot)$ (e.g., frost, gaussian noise, etc.). We analyze the spectrum each corruption $c(\cdot)$ by computing $\frac{1}{N}\sum_{i=1}^{N} PSD(X_i - c(X_i))$ (see Figure 1a). We classify the corruptions into types of frequencies: low, medium, and high using the characterization from [39], which has been adopted by the neural network robustness community. Note that categorizing a corruption as high is more akin to saying it "contains substantial high-frequency content" rather than saying it "exclusively consists of high-frequency content."

## 3 Spectral inspection tools

While existing scalar metrics, such as PSNR, are able to summarize the visual similarity of reconstructed images to the original, we will demonstrate that such metrics can provide an incomplete (and sometimes misleading) picture when measuring the impact of compression in OOD settings. Notably, existing tools do not consider the impact of compression on different frequency ranges of images within a dataset. To more thoroughly analyze the effects of image compression, we propose to measure and visualize the effect of image compression in the spectral domain. Given an image compression model $\mathcal{C}$ that returns reconstructed images, we introduce tools for analyzing compression

---

[2]We used github.com/bethgelab/imagecorruptions to apply corruptions to Kodak and CLIC images

error in the Fourier domain to better understand ($i$) which *spectral frequencies* are distorted by $\mathcal{C}$, ($ii$) the *OOD generalization* error, and ($iii$) the *robustness* error in the presence of distributional shifts.

**Definition 3.1** (Spectral Measure of Distortion Error). To analyze ($i$), we evaluate the image compression model $\mathcal{C}$'s ability to reconstruct components of an image across a range of frequencies. To quantify this, we compute the average PSD of the difference between each image $X_k$ in a dataset $\mathcal{X}$ and the reconstructed version $\mathcal{C}(X_k)$ of $X_k$: $\mathcal{D}(\mathcal{C}, \mathcal{X}) := \frac{1}{N} \sum_{k=1}^{N} PSD(X_k - \mathcal{C}(X_k))$.

**Definition 3.2** (Spectral Measure of OOD Generalization Error). For ($ii$), we evaluate $\mathcal{C}$'s ability to faithfully reconstruct OOD images. To quantify this, we extend the metric $\mathcal{D}(\mathcal{C}, \mathcal{X})$ to account for a corrupted version $c(\mathcal{X})$ of $\mathcal{X}$ as follows: $\mathcal{G}(\mathcal{C}, \mathcal{X}, c) := \frac{1}{N} \sum_{k=1}^{N} PSD(c(X_k) - \mathcal{C}(c(X_k)))$.

**Definition 3.3** (Spectral Measure of OOD Robustness Error). For ($iii$), we evaluate $\mathcal{C}$'s denoising ability. To quantify this, we compute the average PSD of the difference between each uncorrupted image $X_k$ and the reconstructed version $\mathcal{C}(c(X_k))$ of the corresponding corrupted image $c(X_k)$: $\mathcal{R}(\mathcal{C}, \mathcal{X}, c) := \frac{1}{N} \sum_{k=1}^{N} PSD(X_k - \mathcal{C}(c(X_k)))$.

The $PSD$ function used in the above formulas converts the input image from the spatial to frequency domain by applying the Fast Fourier Transform (FFT) followed by a shift operation to center the zero-frequency component. The proposed tools are computed for a set of images by averaging the power spectral density (PSD) of the difference between the compressed image and the original for each image in the set of images. For simplicity, when $(\mathcal{C}, \mathcal{X}, c)$ is clear from the context, we will just write $\mathcal{D}$, $\mathcal{G}$, or $\mathcal{R}$. Note that $\mathcal{G}$ provides insight into the compression model $\mathcal{C}$'s ability to generalize to a distribution shift $c$ while $\mathcal{R}$ visualizes the denoising effect (or lack thereof) of $\mathcal{C}$ across the frequency domain.

In Appendix B, we present results using an additional tool, *the Fourier heatmap*, that utilizes Fourier basis perturbations as corruptions and is used to corroborate our findings for the specific -C datasets and corruptions we consider. This tool can be leveraged when specific OOD data is unavailable.

## 4 Experiments and findings

Using our spectral inspection tools and OOD datasets, we analyze the performance of several image compression methods and identify several inter- and intra-class differences between classic codecs and NIC methods. We show results for two classic codecs, JPEG and JPEG2000, and two NIC models, SH NIC and ELIC, in the main body and report the remaining results in Appendices C and D.

**Classic codecs.** We apply each of the following algorithms over several compression rates $q$.

- **JPEG** [60]
- **JPEG2000** [53]
- Versatile Video Coding (VVC) (equivalently h.266) using the VVC Test Model (VTM) software [30, 3]

**Neural Image Compressors (NIC) Models.** NIC optimization uses a hyperparameter $\lambda$ to control the relative weight of distortion (quality of reconstruction) and rate (level of compression) terms in the objective function. We train each NIC model over several values of $\lambda$. All models were optimized on the train split of the 2020 CLIC dataset [59]. Further details on their model architectures and training can be found in Appendix I.

- **SH NIC**: Scale-hyperprior model from Ballé *et al.* optimized for PSNR (*i.e.,* distortion objective is mean squared error) [9].
- **ELIC**: Efficient Learned Image Compression optimized for PSNR [24].
- Variant of SH NIC optimized for MS-SSIM [9].
- Variable-rate version of SH NIC optimized using Loss Conditional Training [17].
- Variable-rate model above with 80-95% of weights pruned using gradual magnitude pruning [70].

**Evaluation setup.** We compare distortion, robustness, and generalization error of different image compression methods under three constraints: (a) **no constraint**, (b) **fixed-bpp**, and (c) **fixed-PSNR**. In (a), we compare methods over their full range of rate-distortion tradeoffs by generating rate-distortion curves. In (b), we compare models with hyper-parameters which give a very similar bits per pixel (bpp) result on a particular dataset. For example, we find that on the CLIC dataset, SH NIC with

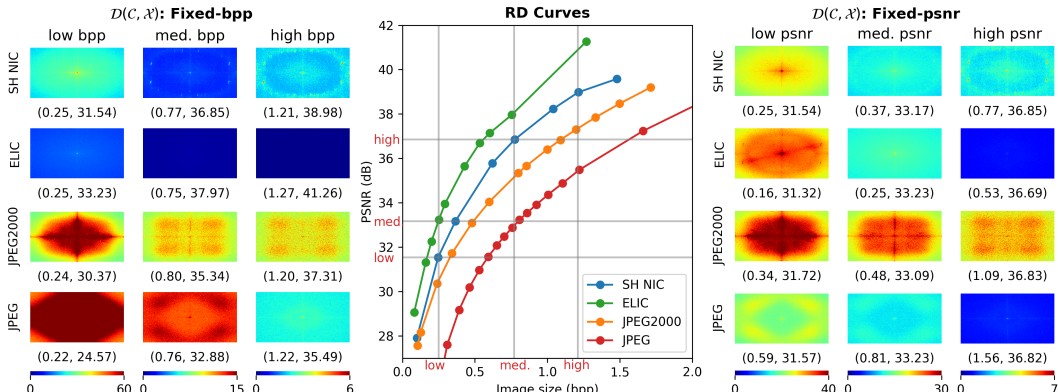

Figure 2: **Visualizing distortion via CLIC test set evaluation. Left:** spectral measure of in-distribution reconstruction error $\mathcal{D}$ under the fixed-bpp constraint at three rates. **Center:** Rate-distortion curves with vertical lines indicating fixed-bpp values and horizontal lines indicating fixed-PSNR values. **Right:** $\mathcal{D}$ under fixed-PSNR constraint. Each $\mathcal{D}$ plot is labeled with a tuple of that model's (bpp, PSNR) on CLIC. Hotter colors (red) indicate more error in that frequency range.

$\lambda = 0.15$, ELIC with $\lambda = 0.15$, JPEG2000 with $q = 10$, and JPEG with $q = 75$ all give a bpp close to 1.21. Thus, comparing these three models with those hyper-parameters on CLIC under a fixed-bpp constraint, *emulates a setting in which a fixed budget is available to store images*. Analogously, in (c) we compare models with hyper-parameters yielding a fixed PSNR. This allows the comparison of spectral properties of data distributions that *achieve the same quality according to the scalar PSNR metric*. Scenarios (b) and (c) are used when evaluating $\mathcal{D}, \mathcal{G}, \mathcal{R}$, Fourier heatmaps, and accuracy on a downstream task.

**Test data.** All models are tested on (a) in-distribution (IND) and (b) corrupted (or OOD) datasets. For (a), we use the 2020 CLIC test split, the full Kodak dataset, and the ImageNet validation split. For (b), we use the corresponding -C datasets for each of the datasets in (a). The main body contains results for the CLIC/CLIC-C dataset. Analogous results for Kodak are in Appendix G.

## 4.1 Evaluating spectral distortion on IND data

On in-distribution (IND) data, the existing RD curve metrics in the center of Figure 2 highlight the established trend that these NIC models outperform the JPEG2000 and JPEG across the compression rates that the NIC model is trained on (bpp $\in [0.1, 1.5]$), with ELIC outperforming SH NIC and JPEG2000 outperforming JPEG.

Next, we use our spectral inspection tool $\mathcal{D}$ to better understand the effects of different image compression methods. Specifically, Figure 2 shows plots of $\mathcal{D}$ under three fixed-bpp and three fixed-PSNR scenarios on the clean CLIC dataset. We highlight some surprising insights below.

**Two methods yielding the same PSNR can produce very different spectral artifacts.** Under the fixed-PSNR constraint (right side of Figure 2), each column consists of methods with hyper-parameters selected to give very similar PSNRs on the CLIC test set (*e.g.*, models on the "high psnr" column all have PSNR $\approx 36.8$). Despite having comparable PSNRs, the plots of $\mathcal{D}$ vary greatly between the four models. In particular, the SH NIC models distort high frequencies more than medium frequencies (notice the warmer-colored rings around the edges of the $\mathcal{D}$ plots with cooler-colored centers). JPEG2000, on the other hand, distorts low and medium frequencies more than high frequencies (notice the large rectangles of warmer colors). ELIC distorts all frequencies relatively evenly (the color gradients in the plots are very narrow).[3] JPEG severely distorts one ring, but this ring includes lower frequencies (has a smaller radius) than the rings left by SH NIC. This same pattern holds under the fixed-bpp constraint (left side of Figure 2). These observations demonstrate that *PSNR is not a comprehensive metric*.

---

[3]We provide a more detailed view of the ELIC plots with smaller color bars in Figure 14 of the Appendix.

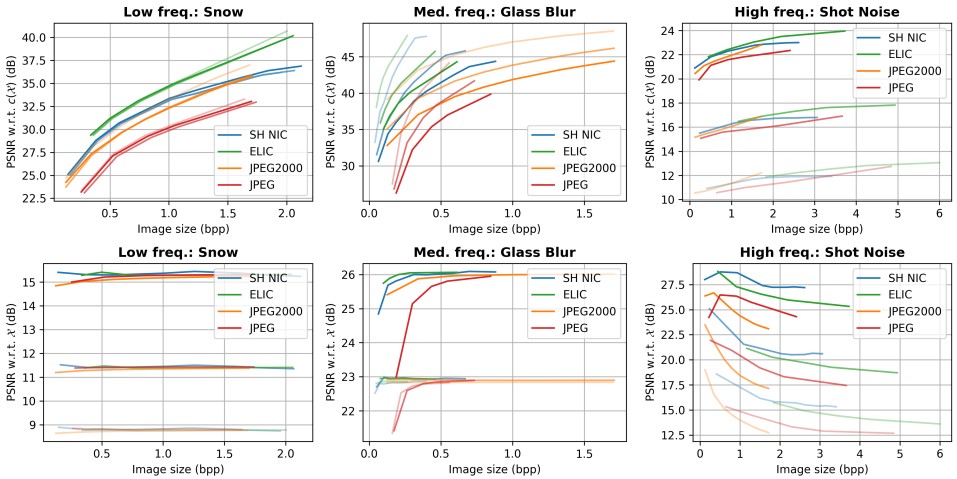

Figure 3: **Rate-distortion curves for a representative low, medium, and high-frequency shift.** Each shift and model has three curves for severity=1 (least transparent), severity=3, and severity=5 (most transparent). **Top row:** generalization of $\mathcal{C}(c(\mathcal{X}))$ w.r.t. $c(\mathcal{X})$ (*i.e.*, PSNR of the reconstructed shifted images w.r.t. the original shifted images). **Bottom row:** denoising of $\mathcal{C}(c(\mathcal{X}))$ w.r.t. $\mathcal{X}$ (*i.e.*, PSNR of the reconstructed shifted images w.r.t. the original clean images).

**As the compression rate increases, different codecs prioritize different parts of the spectrum.** On the left side of Figure 2, each column represents a different "budget" scenario where the three methods have hyper-parameters which result in the models giving very similar bpps on the CLIC test set. Although it was previously known that the quality of the reconstructed images decreases as the bpp decreases, it was not previously known *which* frequencies NIC models distort to achieve a given bpp. The $\mathcal{D}$ plots show that JPEG2000 models corrupt low- and mid-frequency regions starting at low compression rates and these regions become more severe as the budget decreases. SH NIC models do almost the opposite—they sacrifice the highest frequencies first and expand this region into lower frequencies at more severe compression rates (*i.e.*, as bpp decreases). JPEG has a mechanism more similar to JPEG2000 in that it corrupts one ring and increases the severity of this ring, while ELIC models corrupt all frequencies evenly and then corrupt low frequencies more as bpp decreases. This suggests that *as the compression rate increases, classic codecs corrupt the same frequencies more severely while NIC models change their corruption patterns*.

### 4.2 Evaluating the generalization and the robustness on OOD data

We use the CLIC-C dataset and our spectral tools to study the OOD performance of different image compression methods. We show results for example shifts (one low, medium, and high-frequency representative) and three severities (1, 3, and 5 where 5 is most severe) in Figures 3 and 4 and discuss several interesting findings below.

#### 4.2.1 Rate-distortion curves on OOD data

**Image compression models generalize to low- and mid-frequency shifts better than high-frequency shifts.** The top row of Figure 3 shows how well different compression models generalize to shifted images in terms of RD curves. In other words, these plots show how well a compressor $\mathcal{C}$ can reconstruct a given shifted image $c(\mathcal{X})$ in terms of PSNR of $\mathcal{C}(c(\mathcal{X}))$ with respect to $c(\mathcal{X})$. The three examples of corruption in this figure show vastly different trends. On the low-frequency corruption (snow), all four models can reconstruct $c(\mathcal{X})$ almost as well as these models can reconstruct clean images: note the PSNR range in the top left plot of Figure 3 is about 22-40 while the PSNR range for the clean data in Figure 2 is about 28-42 over the same bpp range. Interestingly, the three models can reconstruct the images with the glass blur (a medium-frequency shift) *better* than they can reconstruct clean images (the PSNR of the data shifted with glass blur ranges from about 25-48 with bpp < 1). These results suggest that *image compression models are fairly effective at generalizing to low-frequency shifts and very effective at generalizing to medium-frequency shifts*. However, the

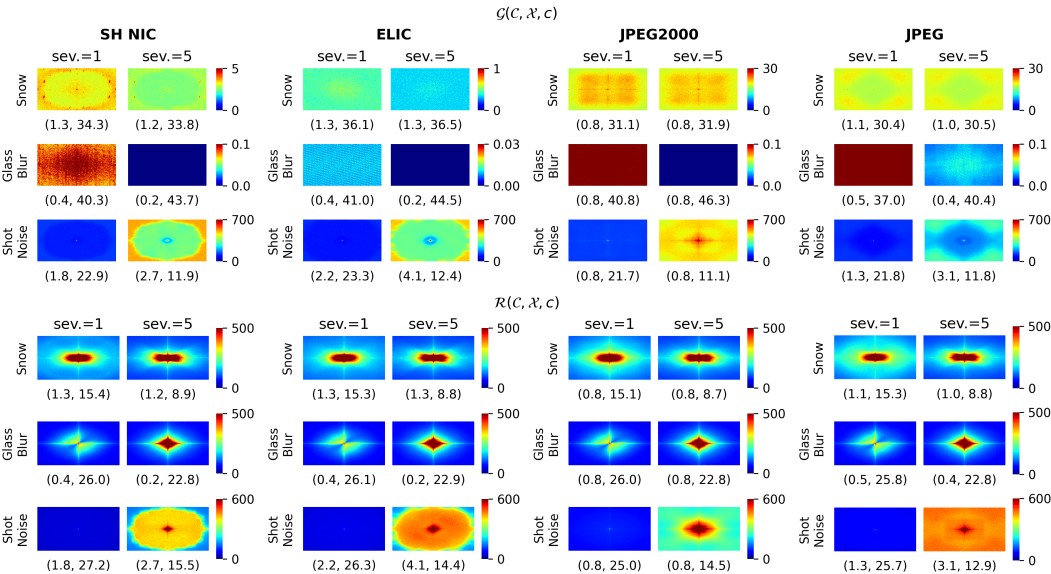

Figure 4: **Generalization error $\mathcal{G}$ (top) and denoising error $\mathcal{R}$ (bottom).** We plot both spectral metrics for one low, medium, and high-frequency corruption at severities 1 and 5. Each plot is labeled with a tuple of that model's (bpp, PSNR) on the CLIC-C dataset with that corruption.

high-frequency shift (shot noise), gives a starkly different result. All four models give very low PSNRs with respect to $c(\mathcal{X})$. Even at the lowest severity (severity=1), this PSNR is only in the low 20s. As the severity increases (*i.e.*, as the lines become more transparent), the PSNR decreases even more to the point that, at the highest severity, none of the models can achieve a PSNR higher than 14. Notably, the main factor determining the PSNR is the severity of the corruption and not the compression model or bpp. This suggests that *it is significantly harder to generalize to high-frequency data than to low- or mid-frequency data*.

**NIC models are better at denoising high-frequency corruptions than classic codecs.** The second row of Figure 3 shows how well different compressors $\mathcal{C}$ denoise corrupted images in terms of the PSNR of $\mathcal{C}(c(\mathcal{X}))$ with respect to $\mathcal{X}$. These results show that all the models fail at denoising snow (low-frequency) and glass blur (medium-frequency) corruptions (PSNR does not change much with an increase in bpp, except at low bpps on glass blur). However, on shot noise (a high-frequency corruption) both NIC models achieve better PSNRs with respect to $\mathcal{X}$ than both classic codecs. This trend is consistent for all three severity levels and suggests that *NICs may be a more effective method for denoising high-frequency corruptions than the previously-used JPEG and JPEG2000 methods*. The implication of this finding extends to the research area of adversarial example denoising [6].

### 4.2.2 Spectral analysis on OOD data

We now take a deeper look at the Section 4.2.1 findings using our spectral inspection tools.

**Spectral artifacts are similar for low-frequency shifts and clean images.** The patterns of $\mathcal{G}$ in Figure 4 measure how well each model generalizes to (or reconstructs) the shifted images. For the low-frequency shift (top row of Figure 4), the plots look strikingly similar to the patterns exhibited by the same models on clean data (Figure 2 top and bottom row): SH NIC models distort high frequencies more than low frequencies while JPEG2000 distorts low and medium frequencies more than high frequencies. Similarly, ELIC distorts all frequencies relatively evenly and JPEG severely distorts one ring of frequencies. This suggests that *the methods' modus operandi for generalization to data with low-frequency shifts is similar to their modus operandi on clean data*, which makes sense as clean data is dominated by low/mid frequencies. Interestingly, these generalization differences between compressors are not accompanied by differences in robustness metric $\mathcal{R}$. All methods show similar patterns in their plots of $\mathcal{R}$ patterns and these in turn look similar to the snow corruption plot in Figure 1a. This is consistent with our finding from Figure 3 which showed that both NIC and

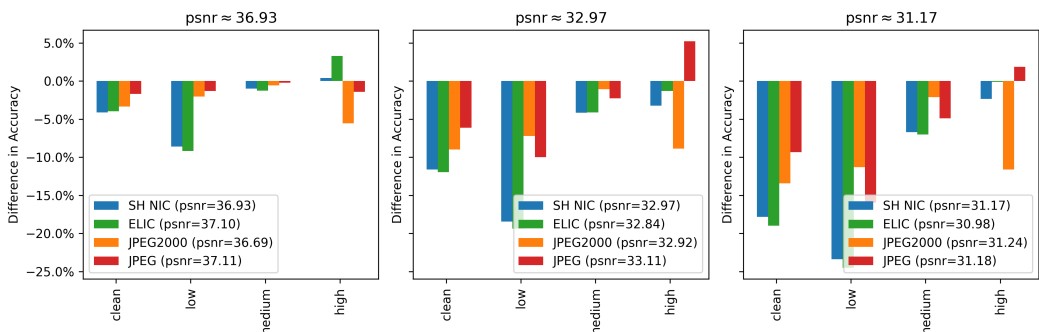

Figure 5: **Effect of compressing corrupt images on classification accuracy.** Each bar shows the average difference in accuracy after compressing with different methods $\mathcal{C}$ for a group of corruptions as classified in Table 1b. Specifically, let $A(X)$ be the top-1 accuracy of the model on dataset $X$, measured in percentage points. Then we report $A(\mathcal{C}(c(X))) - A(c(X))$ over all -C corruptions in the corruption category with severity 3 (or on clean ImageNet in the case of "clean"). Each subplot shows results under a different fixed-PSNR constraint based on the PSNRs achieved by the compressors on the clean ImageNet dataset. Results for individual corruptions are in Figure 15.

JPEG200 fail to denoise low-frequency corruptions. In other words, Figure 4 shows that *NIC and classic codecs fail in a very similar manner on low-frequency signal denoising tasks.*

**Both NIC and classic codecs make almost no generalization error on medium-frequency shifts.** The second row of Figure 4 shows that all four methods have very small generalization errors (magnitudes < 0.2), and this low error is relatively uniform across all frequencies. This shows that all the models models are very effective at reconstructing all the frequencies in images with glass blur—in fact, they can reconstruct these images *better* than they can reconstruct clean images— corroborating our first finding in Section 4.2.1. Again the $\mathcal{R}$ plots for glass blur for both of these models look very similar to Figure 1a; *this similarity has a simple explanation due to a fundamental tradeoff between generalization and robustness.* This relationship between $\mathcal{R}, \mathcal{G}$ and average PSD of corruptions is described more precisely in Appendix L.4.

**High-frequency corruptions uncover the spectral bias of NIC models.** Section 4.2.1 highlighted that high-frequency signals severely degrade the generalization performance of all image compression methods. From the RD curves with respect to the corrupt images (top right plot in Figure 3), we observe that at each severity the three models have almost identical performance in the usual 0-2 bpp range. *These results might lead us to expect that these models make similar reconstruction mistakes, but our spectral inspection tools indicate that this is not the case at all.* Our plots of $\mathcal{G}$ on shot noise at severity=5 (bottom row, columns 2, 4, 6, and 8 of Figure 4) indicate that NIC models distort the highest frequencies significantly more than the low and medium frequencies (notice the orange borders of the plots) while JPEG and JPEG2000 exhibit different patterns. This nuance, which is equivalent to one we previously observed with clean data in Figure 2, becomes more apparent from the plots of $\mathcal{G}$ on a high-frequency corruption because high-frequency signals are much more prevalent on this data than on the clean data. Additionally, the bottom right plot of Figure 3 shows that for shot noise, NIC models achieve higher PSNR of the reconstructed corrupt images with respect to the clean images than the classic codecs, *i.e.,* they have a stronger denoising effect. In the bottom right of Figure 4 we see a more detailed picture: for high severity shot noise, NIC models achieve lower $\mathcal{R}$ in high frequencies. Also, note that the NIC models have the smallest errors in $\mathcal{G}$ whereas they have the largest errors in $\mathcal{R}$ and vice versa. Thus, these findings suggest that *NIC models behave similarly to a low-pass filter.*

### 4.3 Impact of spectral artifacts on downstream applications

In practice, image compression algorithms may be used as a pre-processing step before images are used for another downstream task. For this reason, practitioners should also consider the performance on downstream tasks when comparing compression methods. We analyze the effectiveness of the compression methods on the downstream task of ImageNet classification using a pre-trained ResNet-50 model and report the difference in top-1 accuracy after compression (Figure 5) [25].

**NIC can improve the robustness of downstream classification to high-frequency corruptions.**
While in general, compression degrades classification performance on clean images (all compressors
show negative differences for the "clean" category in Figure 5), in some cases NIC and JPEG
can actually improve the robustness of the classification model against high-frequency corruptions.
Specifically, at PSNR=36.93, the difference in accuracy is *positive* for high-frequency corruptions
with both SH NIC and ELIC, meaning that the classification model gave a higher accuracy on the set
of corrupted images after compression than it did on the original corrupted images. Compressing
with JPEG or JPEG2000 at the same rate caused a degradation in accuracy on these corruptions.
However, at higher compression rates (PSNR=32.97 and PSNR=31.17), JPEG improves classification
performance for high-frequency corruptions, while SH NIC and ELIC do not. On low- and mid-
frequency corruptions, JPEG and JPEG2000 have a smaller degradations in accuracy compared to
NIC the NIC models and no model gives an improvement in classification accuracy. Thus, *the ideal
compressor for downstream tasks is dependent on the type of corruption and level of compression*.

**Pruning NIC models amplifies the robustness gains of NIC for downstream classification tasks.**
Additional experimental results in Appendix C (Figure 9), show that *pruned NIC and NIC optimized
for MS-SSIM act as even better high-frequency signal denoisers for this application*.

## 5 Theoretical analysis of the OOD performance of NIC

We corroborate our empirical findings with theoretical results. Here we summarize theoretical results
on linear autoencoder-based NIC methods. For complete definitions, additional references, and proofs
of the following statements, please refer to Appendix L. In Appendix L.3, we provide a more general
result applicable to nonlinear models.

Recall a classical observation: in the setting of linearly auto-encoding a mean-centered distribution,
the reconstruction function (*i.e.*, encoder-decoder composition) is a projection onto the high-variance
principal components of the input data [18] (see also [7]). Combining this with well-known facts
about statistics of natural images, we show that a linear autoencoder applied to a natural image
dataset retains only low-to-mid (spatial) frequency components, which account for the majority of the
variance. Using this result, we state theoretical explanations for multiple trends in our experiments[4].

**Lemma 5.1.** *Let $\mathcal{X}$ be a dataset of natural images and let $\hat{\mathcal{X}}$ denote its (spatial) discrete Fourier
transform. Assume the following (for supporting evidence see Appendix L):*

- *(I) The principal components of $\hat{\mathcal{X}}$ are roughly aligned with the spatial Fourier frequency basis.*
- *(II) The associated variances are monotonically decreasing with frequency magnitude (more
specifically according to the power law $\frac{1}{|i|^\alpha + |j|^\beta}$).*

*If $\mathcal{C}$ is a linear autoencoder trained by minimizing MSE on $\mathcal{X}$, with latent space of dimension $r$, and
if "$\widehat{\phantom{m}}$" denotes the spatial discrete Fourier transform, for any data point $X$ with Fourier transform $\hat{X}$*

$$\widehat{\mathcal{C}(X)} \approx \begin{cases} \hat{X}_{:,ij} & : i^2 + j^2 \leq \frac{r}{\pi K} \\ 0 & : otherwise. \end{cases}$$

*where $K$ is the number of channels in the images in $\mathcal{X}$ and where $\hat{X}_{:,ij}$ denotes the components of $\hat{X}$
corresponding to spatial frequency $(i, j)$.*

**Corollary 5.2.** *Under the hypotheses of Lemma 5.1, the robustness error of $\mathcal{C}$ to a corruption $c$ (as
defined in Definition 3.3), measured in spatial frequency $(i, j)$, is*

$$\mathcal{R}(\mathcal{C}, \mathcal{X}, c)_{ij} \approx \begin{cases} \frac{1}{N} \sum_k |(\widehat{c(X_k)} - \hat{X}_k)_{:,ij}| & : i^2 + j^2 \leq \frac{r}{\pi K} \\ \frac{1}{N} \sum_k |\hat{X}_{k,:,ij}| & : otherwise. \end{cases}$$

**Corollary 5.3.** *Under the hypotheses of Lemma 5.1, the generalization error of $\mathcal{C}$ to a corruption $c$
(as defined in Definition 3.2), measured in spatial frequency $(i, j)$, is*

$$\mathcal{G}(\mathcal{C}, \mathcal{X}, c)_{ij} \approx \begin{cases} 0 & : i^2 + j^2 \leq \frac{r}{\pi K} \\ \frac{1}{N} \sum_k \sqrt{|\hat{X}_{:,ij}|^2 + 2\hat{X}_{:,ij}(\widehat{c(X)} - \hat{X})_{:,ij} + |(\widehat{c(X)} - \hat{X})_{:,ij}|^2} & : otherwise. \end{cases}$$

---

[4]Since this simplified model has no compression rate objective, one should only expect the above theoretical
results to be predictive (or explanatory) of the high bpp/PSNR cases of our experiments.

Lemma 5.1 suggests that autoencoder compressors trained on natural images behave like low-pass filters, corroborating our claim in Section 4.2.2 (see also remark L.1).

Corollary 5.2 suggests that *autoencoder compressors trained on natural images are less robust to corruptions with large amplitude in low frequencies* (in the sense that $|(c(X) - X)_{:,ij}|^2$ is large for small values of $ij$). This is indeed what we see in Figure 3 (bottom left), where snow corruptions are detrimental to PSNR of $\mathcal{C}(c(\mathcal{X}))$ w.r.t. $\mathcal{X}$, and Figure 4, where the $\mathcal{R}(\mathcal{C}, \mathcal{X}, c)$ error for the NIC is concentrated in low frequencies. We also observe in Figure 5 that NIC is more beneficial for downstream classification accuracy in the case of high-frequency corruptions (e.g. shot noise) and less beneficial in the case of low-frequency corruptions (e.g. snow).

On the other hand, the conclusion of Corollary 5.3 is more involved than that of Corollary 5.2– the "cross term" $2\hat{X}_{:,ij}(\widehat{c(X)} - \hat{X})_{:,ij}$ is in general non-zero. However, there are cases where in expectation over the data set $\mathcal{X}$ this cross term vanishes (*e.g.*, when $c$ is additive noise). In such cases, Corollary 5.3 suggests that *compressors trained on natural images generalize less successfully to shifts with large amplitude in high frequencies*. In Figure 3 (top right) we see that shot noise corruptions are detrimental to PSNR of $\mathcal{C}(c(\mathcal{X}))$ w.r.t. $c(\mathcal{X})$, and in Figure 4 we see that for the snow and shot noise corruptions, $\mathcal{G}(\mathcal{C}, \mathcal{X}, c)$ are concentrated in high frequencies.[5] In summary, with both theoretical analysis of a simple mathematical model and empirical results, we find that *NICs have a spectral bias that causes them to overfit to discard high frequency components of natural images.*.

# 6   Limitations

Due to the number of axes of variation present in our experiments (*e.g.*, model type, compression rate, evaluation dataset, corruption type, corruption severity, evaluation metric), we were forced to constrain many variables. We present results for a limited number of NIC models all trained on one dataset, CLIC. Due to limitations on the number of models we could train and discrete choices for classic codec hyper-parameters, there is some variation within our fixed- {bpp,PSNR} bins.

Evaluating the robustness of any machine learning system is inherently a multi-faceted problem riddled with unknown-unknowns. While we focus on robustness to naturalistic input-data corruptions, there are other important distribution shifts for researchers and practitioners to consider, such as subtle changes in data collection methodology [51]. We omit a study of NIC model *adversarial* robustness to worst-case input data perturbations. Nevertheless, we hope that the our evaluation methods (including our metrics $\{\mathcal{D}, \mathcal{G}, \mathcal{R}\}$ and Fourier heatmap spectral inspection tools) will be useful in future work on NIC model robustness.

# 7   Conclusion and future directions

We proposed benchmark datasets and inspection tools to gain a deeper understanding of the robustness and generalization behavior of image compression models. Using our spectral inspection tools, we uncovered the modus operandi of different compression methods. We also highlighted similarities and differences among them via a systematic OOD evaluation. Exploring the use of our tools in other image compression methods, including transformer-based architectures and implicit neural representations methods is expected to provide interesting insights [41, 72, 73, 45, 54]. Our OOD evaluations identify NIC model brittleness issues: designing practical robust training approaches and/or methods to efficiently adapt to distribution shifts at runtime is a worthwhile research direction.

## Acknowledgements

This work was performed under the auspices of the U.S. Department of Energy by Lawrence Livermore National Laboratory under Contract DE-AC52-07NA27344 and was supported by the LLNL-LDRD Program under Project No. 22-DR-009 (LLNL-JRNL-851532).

---

[5]Moreover glass blur can be viewed as a sort of convolution operator, for which one can verify that the aforementioned cross term is non-zero.

The third author was supported by the Laboratory Directed Research and Development Program at Pacific Northwest National Laboratory, a multiprogram national laboratory operated by Battelle for the U.S. Department of Energy.

We thank Eleanor Byler for helpful comments on an earlier draft.

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

# A Background and Related Work

## A.1 Neural Image Compression (NIC)

In lossy image compression, an input image is compressed into a small file or bitstream for transmission or storage. This file is then used to reconstruct the original image. Several models have been proposed for neural (or learned) image compression such as convolutional neural networks (CNNs), autoencoders, and generative adversarial networks [22, 8, 9, 46, 58, 13, 65, 71, 47]. In the most basic version of these models, an image vector $x$ is mapped to a latent representation $y$ using a convolutional autoencoder. $y$ is then quantized into $\hat{y}$, so it can be compressed to a bitstream via an entropy coding method, such as arithmetic coding [52]. During reconstruction, the received bits go through the reverse entropy coding process and the revealed $\hat{y}$ is fed into a CNN decoder which outputs the reconstructed image $\hat{x}$. These models can be trained end-to-end by using differentiable proxies of entropy coding [8].

Normally entropy coding relies on one prior probability model of $\hat{y}$, which is learned from the entire set of images. Ballé et al. [9], however, recognized that the prior of this model can be optimized for each image separately. By adding an additional hyperprior autoencoder to their model, they learn the latent representation for the entropy model of each image. The information transmitted from this autoencoder can be seen as equivalent to "side information" used by other codecs. Minnen et al. [46] further improved this model by adding an autoregressive prior to the hierarchical prior used by Ballé. The Efficient Learned Image Compression (ELIC) model builds on these models by adding uneven channel-conditional adaptive coding and a spatial context model [24].

## A.2 Generalization and robustness in DNNs for classification

It is well known that deep neural networks (DNNs) are prone to overfitting to the specific data distribution they are trained on. Much recent work has focused on improving the robustness of DNNs to other distributions of data, including common image corruptions [27, 66]. CIFAR-10-C and ImageNet-C were proposed as a benchmark datasets to test the robustness of models to common image corruptions such as corruptions from weather, blur, and digital noise [26]. Methods for improving robustness include data-level techniques such as adding augmentations and algorithm-level techniques like using larger models or adversarial training. [27, 32, 28, 14].

Yin *et al.* study the effects of data augmentation and adversarial training on different corruptions using a Fourier analysis [66]. They analyze the CIFAR-10-C dataset and find that most augmentation methods help in the high-frequency domain, but hurt in the low-frequency domain. However, it is unknown how image compression affects corrupted images. We analyze how different types of image compression affect the Power Spectral Density (PSDs) of both clean and corrupted data.

## A.3 Generalization and robustness with NIC models

There is a limited amount of work studying the generalization and robustness of NIC models. [37] formulates a problem for designing robust NIC models and train DNNs to be robust to two particular types of distributional shifts. They find a tradeoff between OOD robustness and IND performance. Our work is complementary to this work in that we provide additional metrics and OOD benchmarks to analyze the robustness of various models in more detail. [42] studies NIC models' robustness to white-box and black-box adversarial attacks and proposes new models which improve the robustness of NIC to such attacks. Our work instead looks at naturalistic distribution shifts, which is another important aspect of robustness [26].

[2] designed a NIC module which works on a specific set of OOD images. In particular, their module works on high dynamic range (HDR) images, despite being trained on mostly standard dynamic range (SDR) images. Although this is a specific example of testing NICs on OOD data, our work differs from theirs in that we analyze how NICs would perform on a wide variety of image corruptions without seeing any examples of these corruptions during training.

Additionally, [38] addresses the issue of "multi-generational loss", or the problem of a rapid degradation of quality as images are repeatedly encoded and decoded, with NIC models. Our work differs because we propose spectral tools to analyze the effects of one cycle of encoding and decoding and we study how this affects the performance of different downstream tasks. While multi-generation

loss is an important area of study for tasks like image editing, we focus on analyzing the spectral distortions caused by one cycle of image compression and how these affect a model's generalization and robustness to OOD data. Furthermore, [38] proposes using JPEG2000 as a method to improve the robustness of classification tasks by compressing adversarial noise. We explore how NIC models compare to JPEG2000 in terms of their robustness to several common types of image corruptions.

## A.4 Lightweight NIC via model pruning

In an effort to reduce the energy and compute cost required by deep neural networks (DNNs), much work has been devoted towards finding methods to approximate DNNs. Common approaches include pruning network weights or quantizing network weights and activations [19, 20, 48, 69, 40, 15]. Remarkably, large sparse models have been shown to outperform small dense models with the same number of weights [70]. Additionally, approximating DNNs can have benefits in terms of robustness on OOD data [16].

Nonetheless, limited efforts to prune or quantize NIC models have been explored. Recently, Gille et al. [21] successfully introduced structured sparsity into NIC models to simultaneously reduce storage and power costs while maintaining a similar rate-distortion curve when compared to a dense version of the same NIC architecture. Their findings produced $\sim$80% reduction in the number of parameters and $\sim$30% reduction in MAC operations required by the NIC encoder. Yin et al. [68] also explored the use of structured sparsity in NIC and their proposed ABCM method was demonstrated to provide up to 7$\times$ and 3$\times$ reduction in DNN parameters and inference time, respectively, with only a mild reduction to reconstruction quality. Luo et al. [43] proposed a technique for pruning NIC models that was demonstrated to maintain the same bpp as dense models but provided a parameter reduction of at most $\sim$34%. Sun et al. [55] demonstrated a technique for quantizing both NIC weights and activations to an 8-bit fixed point representation while yielding a rate-distortion curve comparable to the original 32-bit float NIC model. Outside of DNN pruning or quantization, alternative techniques have been explored for more efficient NIC [31, 67, 23].

## A.5 Metrics for quantifying image compression quality

The objective of lossy image compression is to achieve a balance between minimizing 1) the size of the compressed file and 2) the distortion of the reconstructed image compared to the original image. As these objectives are inherently at odds, the performance of lossy image compression is typically measured using a rate-distortion curve, which measures the size of the compressed image (rate) versus the quality of the reconstructed image (distortion). The rate is usually measured in bits per pixel (bpp), which is the average number of bits required to store each pixel in the compressed representation. The distortion is usually measured by scalar metrics such as peak signal-to-noise ratio (PSNR) or a visual similarity metric such as MS-SSIM [62].

## A.6 Benchmark datasets

We utilize two benchmark datasets for evaluating performance of our NIC models. For training and testing, we make use of CLIC (Challenge on Learned Image Compression) 2020 [59]. This collection is comprised of 1633 training, 102 validation, and 428 test images. The images are of varying resolution and are further categorized as either professional or mobile. To evaluate generalization, we make use of the Kodak [34] dataset which consists of 24 images of resolution $768 \times 512$ (or $512 \times 768$).

# B Additional results: Fourier heatmaps

While our spectral tools allow us to measure different capabilities of compression models, they require the availability of OOD, or corrupted, data. To support a setting in which such OOD data is unavailable, we propose adopting another tool: the Fourier sensitivity heatmap. This tool evaluates the PSNR of a compression model on data perturbed with Fourier basis elements [66]. The resulting visualization is a heatmap where the value at coordinate $(i, j)$ is the PSNR of the compression method on perturbed data $\{X_k + r_k \varepsilon U_{i,j}\}_{k=1}^N$, where each $r_k$ is selected uniformly at random from $\{-1, 1\}$, $\varepsilon$ is the norm of the perturbation, and $U_{i,j}$ is the $(i, j)$th Fourier basis matrix. From [66],

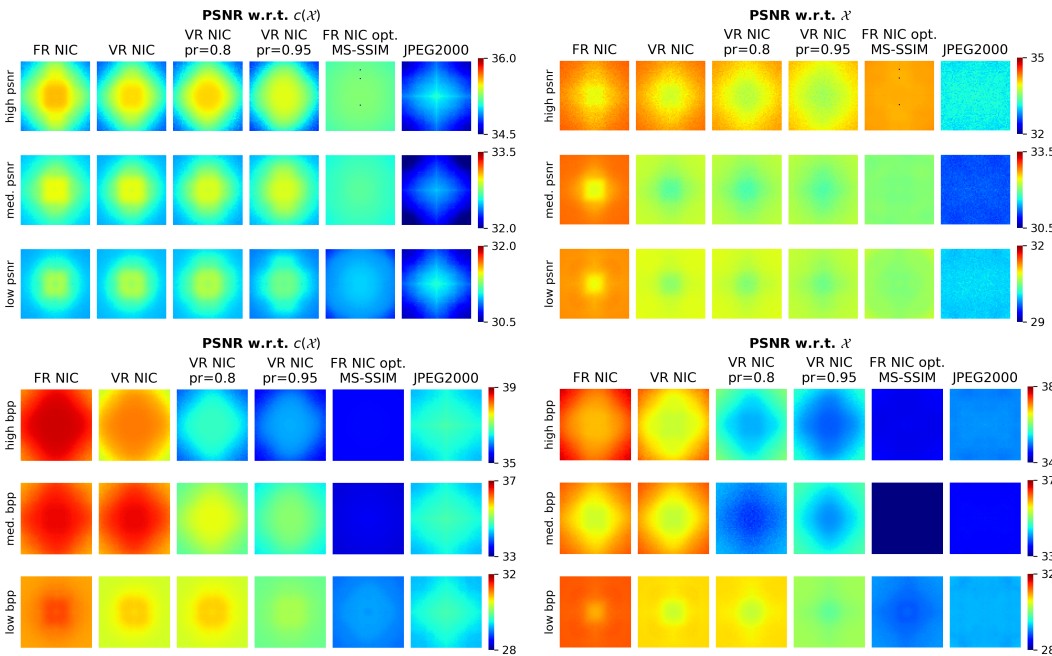

Figure 6: **Fourier heatmaps of methods under the fixed-bpp setting (top) and fixed-PSNR setting (bottom). Left:** PSNRs of $\mathcal{C}(c(\mathcal{X}))$ with respect to $c(\mathcal{X})$. **Right:** PSNR of $\mathcal{C}(c(\mathcal{X}))$ with respect to $\mathcal{X}$. Warmer colors (red) indicate higher PSNR.

$U_{i,j} \in \mathbb{R}^{n_1 \times n_2}$ and satisfies (i) $\|U_{i,j}\| = 1$ and (ii) $\mathcal{F}(U_{i,j})$ has at most two non-zero coordinates specifically at $(i,j)$ and the coordinate symmetric to $(i,j)$ about the matrix center.

We consider two versions of Fourier heatmaps: (a) with respect to perturbed data and (b) with respect to original data. To analyze (a) and (b), the PSNR at each coordinate $(i,j)$ is computed with respect to the perturbed dataset, $\{X_k + r_k \varepsilon U_{i,j}\}_{k=1}^{N}$, and the unperturbed dataset, $\{X_k\}_{k=1}^{N}$, respectively. (a) and (b) can be seen as measures of generalization and robustness respectively.

We computed Fourier heatmaps over the CLIC dataset in Figure 6. We use models from Appendix C, where FR NIC is equivalent to SH NIC in the main body and selected hyper-parameters for these models using a fixed-bpp/PSNR constraint on the clean data. Note: in these Fourier heatmap plots, warmer colors represent *higher PSNR*, which is in contradiction to the plots of $\mathcal{D}$, $\mathcal{G}$, and $\mathcal{R}$ where warmer colors represented *more error*.

On the left side of Figure 6, we analyze the generalization of the image compression models to various frequency shifts. We find that all image compression models generalize to low and medium-frequency signals better than high-frequency signals. Specifically, observe how the centers of the plots–corresponding to low-frequency signals–have hotter colors than the edges of the plots. This corroborates our findings from both the RD curves and $\mathcal{G}$ metrics on the CLIC-C dataset in Section 4.2 and Appendix H. For FR NIC opt. MS-SSIM, this difference is only noticeable at the lowest bpp; however for the other models, there is a clear difference in performance for low and high-frequency shifts at all bpps. Additionally, the effect of pruning is a degradation in PSNR at all frequencies.

On the right side of Figure 6, we analyze the robustness, or denoising, capabilities of the models. These plots show that NIC models are better at denoising high-frequency corruptions than low-frequency corruptions (notice the cooler-colored diamonds in the center of the NIC plots). Meanwhile JPEG2000 has very consistent and poor denoising properties across the entire range of corruption frequencies (the plots are blue with a narrow gradient of color compared to the NIC plots). According to this metric, NIC opt. MS-SSSIM also has poor denoising capabilities (the plots are all dark blue compared to the NIC plots) and pruning NICs also results in a degradation of denoising performance. At first, this may seem like a contradiction to our downstream classification results from Appendix C, which showed that NIC opt. MS-SSIM and VR NIC pr=0.8/0.95 were more effective at denoising high-frequency corruptions than FR NIC and VR NIC. However, recall that each pixel in these plots

only shows the *average PSNR* of the entire perturbed dataset as opposed to the spectral error by frequency, like our $\mathcal{D}$, $\mathcal{G}$, and $\mathcal{R}$ plots show. Then from the main body and Figure 6, it is clear that this PSNR metric is not detailed enough highlight the nuances between the spectral distortion patterns of different methods.

## C   Additional results: NIC variants

In this section, we analyze several additional variants of NIC models. Many of these variants were chosen because they have potential to reduce the amount of memory required to store or run these models, making them more amenable to "in-the-wild" use cases (*e.g.*, the Mars Exploration Rover discussed in the introduction). We also consider models optimized for MS-SSIM because we are interested in how the distortion objective affects the spectral distortion artifacts of the model.

Because all of these models are based on the scale-hyperprior architecture from [9], we use a different naming scheme here than in the main body. Below are descriptions of the NIC models compared in this section:

- **FR NIC:** Equivalent to SH NIC in the main body. Eight fixed-rate models each trained on a unique $\lambda$. Distortion is MSE (*i.e.*, model is optimized for PSNR).
- **VR NIC:** One variable-rate model trained over a continuous range of $\lambda$ values using loss conditional training. Distortion is MSE. More details about this model are provided in Appendix J.
- **VR NIC pr=0.8/0.95:** One variable-rate model trained over a continuous range of $\lambda$ values. Distortion is MSE. Here, gradual magnitude pruning (GMP) is applied during training to prune 80% or 95% of the convolutional and FiLM weights. More details of the GMP algorithm are provided in Appendix K.
- **FR NIC opt.  MS-SSIM:** Eight fixed-rate models, each trained on a unique $\lambda$ value. Distortion is (1 - MS-SSIM) (*i.e.*, model is optimized for MS-SSIM).

Figures 7, 9, and 8 show results for these models. We discuss our findings below.

### C.1   Evaluating spectral distortion on IND data

**Effect of approximating FR NIC with VR NIC**. According to the RD curves in Figure 7, VR NIC obtains the same performance as FR NIC for low and moderate bpps; however, the SH NIC model outperforms the VR NIC model at higher bpps despite being trained on the same range of $\lambda$. This result follows [17] and suggests that the VR NIC may not be expressive enough to learn the high PSNR regime. In terms of the spectral error $\mathcal{D}$, VR NIC and FR NIC have the same error patterns, but VR NIC does leave a slightly higher magnitude of error in some settings (*e.g.*, med. bpp, high bpp, and high psnr).

**Effect of pruning**. VR NIC, VR NIC pr=0.8, and VR NIC=0.95 all have the same VR NIC model architecture, but have 0%, 80%, and 95% sparsity respectively. The RD curves show that pruning causes a degradation in performance, especially for high bpp (see blue lines in Figure 7). However, the PSDs reveal that, even if we make up for the PSNR performance gap by increasing the bpp (right side of Figure 7), the pruned models left a different spectral artifacts from the dense ones. In particular, as the prune rate increases, the ring in the high-frequency region of the PSDs becomes thicker. Presumably reconstructing signals at high-frequencies requires more parameters than reconstructing low frequency signals and so the more sparse models focus on optimizing the "easier" frequencies on the spectrum. These results show that pruning NICs exacerbates the spectral differences between NIC and JPEG2000 that we observed in Figure 2.

**Effect of optimization variable in NIC.** The FR NIC and FR NIC opt. MS-SSIM have the same FR NIC model architecture, but optimize for different "distortion" metrics (*i.e.*, FR NIC optimizes for PSNR while FR NIC opt. MS-SSIM optimizes for MS-SSIM). Unsurprisingly, Figure 7 shows that the FR NIC opt. MS-SSIM model has the worst performance of all the models in terms of RD curves where the distortion metric is PSNR. However, the plots of the distortion error $\mathcal{D}$ give us more insights into the spectral errors that this model is making. Surprisingly, FR NIC opt. MS-SSIM produces spectral artifacts which differ greatly from both the PSNR optimized NIC models and the classic JPEG2000 codec. In particular, like JPEG2000, this model distorts the medium and low frequencies more than high-frequencies (notice the cooler-colored ring around the hot-colored center).

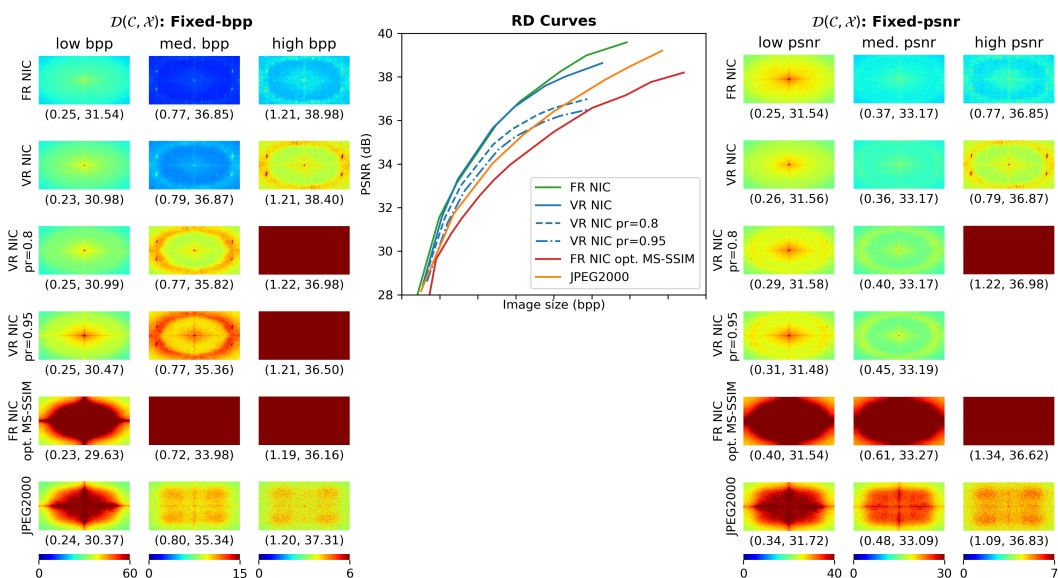

Figure 7: **Visualizing distortion via CLIC test set evaluation with NIC variants. Left:** spectral measure of in-distribution reconstruction error $\mathcal{D}$ under the fixed-bpp constraint at three rates. **Center:** Rate-distortion curves with vertical lines indicating fixed-bpp values and horizontal lines indicating fixed-PSNR values. **Right:** $\mathcal{D}$ under fixed-PSNR constraint. Each $\mathcal{D}$ plot is labeled with a tuple of that model's (bpp, PSNR) on CLIC. Hotter colors (red) indicate more error in that frequency range.

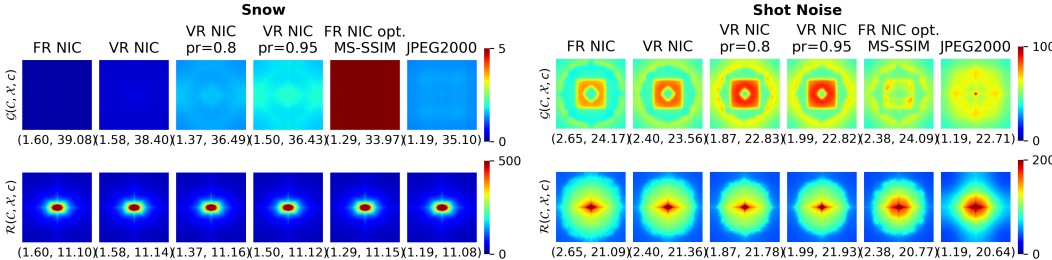

Figure 8: **Spectral measures $\mathcal{G}$ and $\mathcal{R}$ for all NIC variants on two example corruptions of ImageNet-C.** Model hyper-parameters were chosen such that they give bpp$\approx$1.23 on ImageNet. Each plot is labeled with the model's bpp and PSNR with respect to $c(\mathcal{X})$ (top row) or $\mathcal{X}$ (bottom row).

However, it differs from JPEG2000 in that it leaves a less rectangular artifact. Thus, the optimization objective of NIC can influence the spectral artifacts and should be thoroughly examined before being used in practice.

## C.2 Robustness and generalization

We analyze the findings further using our robustness spectral inspection tools on ImageNet-C in Figure 8. The plots of $\mathcal{G}$ on the snow corruption show that although the models have different magnitudes of errors, all models leave similar magnitudes of errors across the frequency spectrum (*i.e.*, the gradient of each plot is narrow). Furthermore, all models have similar plots of $\mathcal{R}$. This follows the result of the next section (accuracy on the downstream task), which shows that all six models have comparable accuracies at this compression rate (see snow column of bpp$\approx$1.23 plot in Figure 9).

On the shot noise corruption, however, we see some notable differences between models on both $\mathcal{G}$ and $\mathcal{R}$. On $\mathcal{G}$, we see that as the prune rate increases, the ring of high-frequency spectral error becomes significantly more bold. This reiterates our finding from Figure 7 that pruning results in a degradation of high-frequency signal reconstruction. Similarly, FR NIC opt. MS-SSIM has a thicker

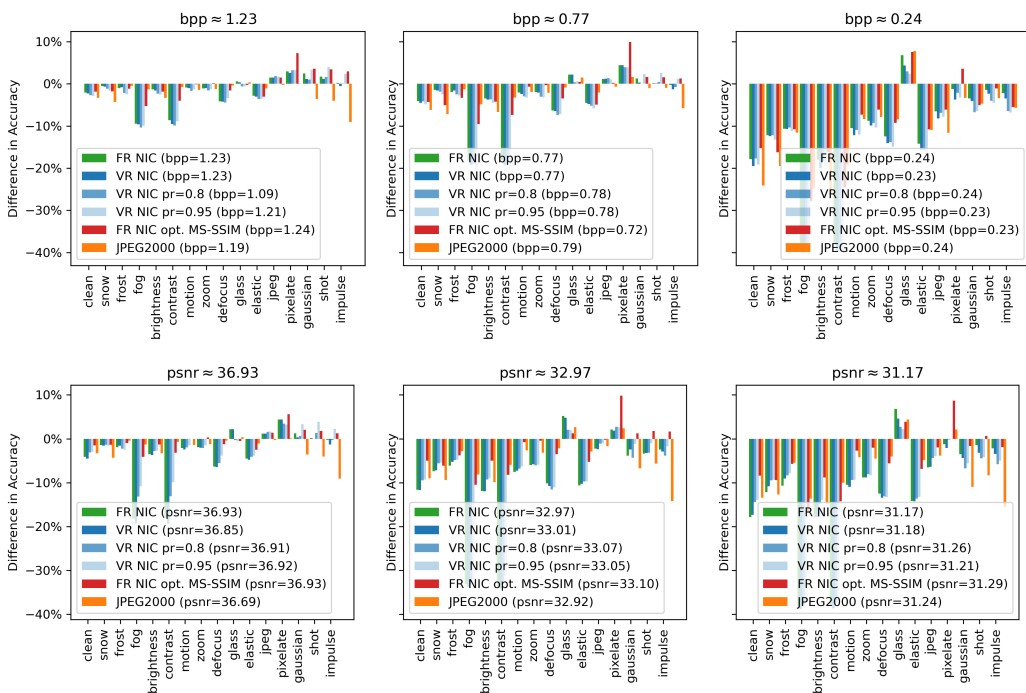

Figure 9: **Effect of compressing corrupt images on classification accuracy.** Each subfigure shows results under a different fixed-bpp or fixed-PSNR constraint based on the bpp/PSNRs achieved by the compressors on the clean ImageNet dataset. Specifically, let $A(X)$ be the top-1 accuracy of the model on dataset $X$, measured in percentage points. We report $A(\mathcal{C}(c(X))) - A(c(X))$ over all -C corruptions (or on clean ImageNet in the case of "clean").

ring than FR NIC in the high-frequency domain. Interestingly, this model also has a smaller error in the low frequency domain compared to the NIC models optimized for PSNR. Finally, JPEG2000 distorts low- and medium-frequencies relatively evenly and distorts high-frequency signals less than NIC which follows from Section 4.1. We can see the effect of these differences in $\mathcal{G}$ in the high-frequency domain of of the plots $\mathcal{R}$. Specifically, the VR NIC leaves a slightly smaller artifact than FR NIC (*i.e.*, the circle of warm colors is larger with VR NIC). This differences is exacerbated as the prune rate increases (*i.e.*, NIC pr=0.95 has the smallest artifact of the four first four models), which suggests that the pruned NIC models are most effective for denoising high-frequency corruptions. FR NIC opt. MS-SSIM also leaves a smaller artifact than FR NIC; however, this has a different shape than the NIC models optimized for PSNR (specifically, there are jagged edges on the edges of this artifact). This suggests that FR NIC opt. MS-SSIM denoises high-frequency data better than FR NIC, but in a different way from VR NIC pr=0.8/0.95. Finally, JPEG2000 has diamond-shaped artifact which suggests it denoises some high-frequency signals, but not all.

### C.3 Downstream task

We also consider how these NIC variants perform on the same downstream classification task outlined in Section 4.3. Figure 9 shows the difference in accuracy before and after corruption for each model on each corruption (or clean data) and Appendix M has tables with these accuracies. We find that the pruned models are better at denoising high-frequency corruptions than the FR NIC. In particular, we find that on the high-frequency corruptions at bpp = 1.23 and 0.77, the pruned VR NICs show a *larger improvement in accuracy* than the dense VR and FR NICs. This finding matches our observation in Figure 7 that the pruned NICs denoise high-frequency signals even more than dense NICs, who in turn denoise these signals more than JPEG2000. The pruned models perform comparably to the dense models on low- and medium-frequency corruptions. Similarly, we also find that NIC opt. MS-SSIM also improves performance on the downstream task on high-frequency corruptions over (a)

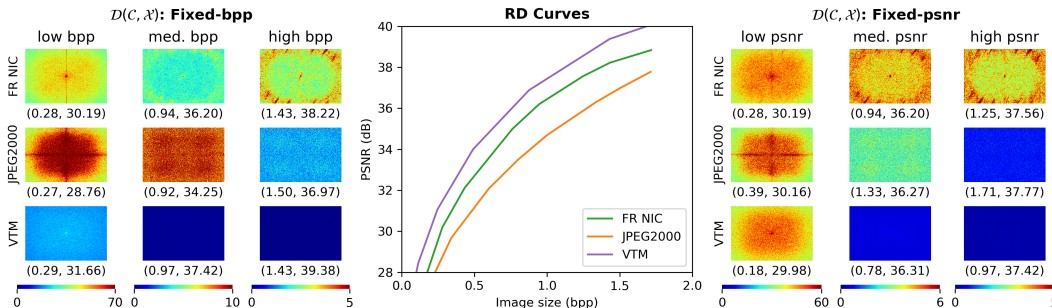

Figure 10: **Visualizing distortion of VTM via Kodak test set evaluation. Left:** spectral measure of in-distribution reconstruction error $\mathcal{D}$ under the fixed-bpp constraint at three rates. **Center:** Rate-distortion curves with vertical lines indicating fixed-bpp values and horizontal lines indicating fixed-PSNR values. **Right:** $\mathcal{D}$ under fixed-PSNR constraint. Each $\mathcal{D}$ plot is labeled with a tuple of that model's (bpp, PSNR) on Kodak. Hotter colors (red) indicate more error in that frequency range.

no compression and (b) compression with dense NIC models. This model also has better performance than the other NIC variants and low- and medium-frequency corruptions.

# D  Additional results: comparing JPEG2000 and VTM

In this section, we compare two classic codecs: JPEG2000 and Versatile Video Coding (VVC) (also known as h.266) codec [30]. We refer to the latter codec as VTM because we used VVC Test Model (VTM) software from [64] to test it. Although VTM has state-of-the-art performance for classic codecs, we chose to use JPEG2000 as the representative classic codec in the main body for three reasons.

1. **JPEG2000 has more consistent bpp usage.** As shown later in this section (Figure 12), VTM varies its bpp usage drastically depending on the corruption type and severity. Specifically, a hyper-parameter which gives a bpp of $x$ on clean data may give a bpp of over $7x$ on certain OOD shifts and severities.[6] Thus, it was unclear how to fairly compare VTM and NIC in a fixed-bpp setting on OOD data.
2. **JPEG2000 implementation has faster inference time.** The current open-source implementation for VTM is significantly slower than those available for JPEG2000 (10s to 100s of seconds for VTM vs. < 1 second for JPEG2000). Thus, JPEG2000 was a more feasible codec for our extensive experimentation.
3. **After adjusting for PSNR, VTM shows similar spectral results as JPEG2000.** Finally, although there are differences in magnitudes of the spectral distortion plots, JPEG2000 and VTM exhibit similar patterns, so we found it sufficient to use only one classical codec representative.

We include results comparing JPEG2000 and VTM on the Kodak and Kodak-C datasets in Figures 10-13 and explain our findings below.

## D.1  Evaluating VTM on IND data

Figure 10 is analogous to Figure 2 in the main body. From the RD curves, we see that VTM outperforms FR NIC and JPEG2000 across all tested bpps. VTM's plots of $\mathcal{D}$ are most similar to JPEG2000's. Specifically, at low compression rates (high bpp/PSNR), both methods leave similar errors across all frequencies (notice the small gradient of colors in both JPEG2000 and VTM's plots of $\mathcal{D}$ at med/high bpp and med/high PSNR). Similarly, at high compression rates (low bpp/PSNR) both models leave more spectral errors at the low and medium frequencies. Despite having similar patterns to JPEG2000, most of the VTM plots have a smaller magnitude than JPEG2000, even at fixed-PSNRs. This suggests that the metric $\mathcal{D}$ metric is more closely aligned with VTM's modus operandi than JPEG2000's.

---

[6]For example, when q=17, VTM uses 2.32 bpp on Kodak and 16.49 bpp on Kodak-C shot noise with severity=5.

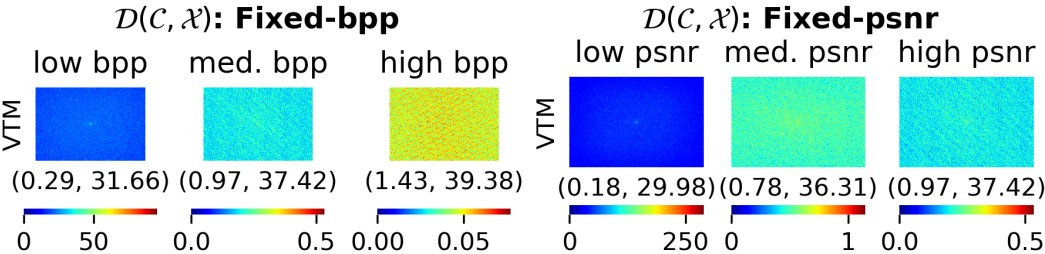

Figure 11: Unscaled plots of $\mathcal{G}$ and $\mathcal{R}$ with VTM from Figure 10.

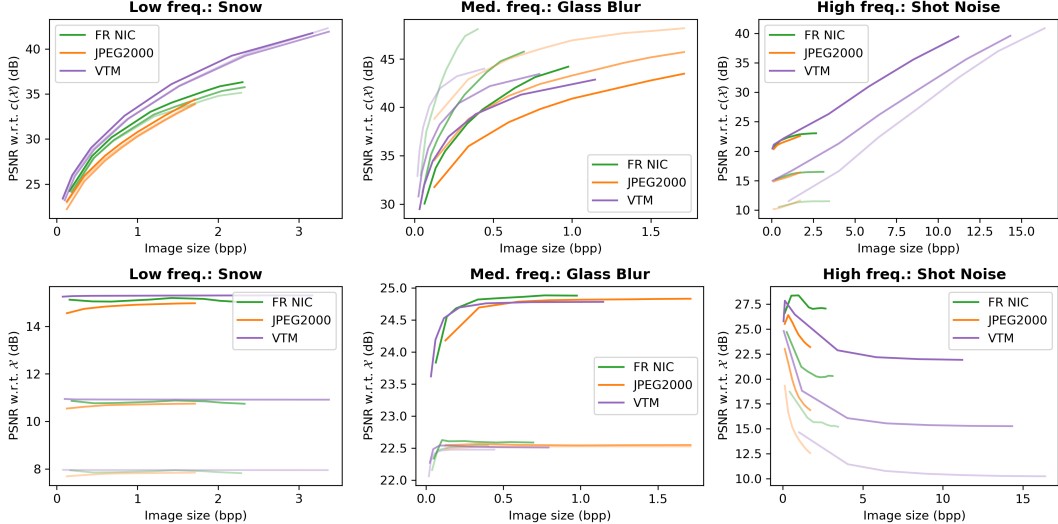

Figure 12: **Rate-distortion curves for a representative low, medium, and high-frequency shift.** Each shift and model has three curves for severity=1 (least transparent), severity=3, and severity=5 (most transparent). **Top row:** generalization of $\mathcal{C}(c(\mathcal{X}))$ w.r.t. $c(\mathcal{X})$ (*i.e.*, PSNR of the reconstructed shifted images w.r.t. the original shifted images). **Bottom row:** denoising of $\mathcal{C}(c(\mathcal{X}))$ w.r.t. $\mathcal{X}$ (*i.e.*, PSNR of the reconstructed shifted images w.r.t. the original clean images).

Figure 11 shows the six VTM plots from Figure 10 without constrained color map scales. These results confirm that VTM leaves only a small magnitude of error, which is even across frequencies.

## D.2 Evaluating VTM on OOD data

Figure 12 shows the RD curves of VTM on a low, medium, and high-frequency shift. On the low-frequency shift (snow), VTM has comparable performance to FR NIC and JPEG2000 at low bpp; however, VTM can achieve higher PSNRs by increasing its bpp usage. This finding suggests that VTM, like the other codecs in Section 4.2.1, generalizes well to low-frequency shifts. However, this highlights the fact that VTM may drastically increase its bpp usage on OOD data. Interestingly, all three models are comparably robust to the snow corruption (bottom row of Figure 12) and the severity of the corruption is the largest determinant of the PSNR with respect to $\mathcal{X}$.

On the medium-frequency shift (glass blur), VTM, like the other codecs in Section 4.2.1, achieves PSNRs which are higher than the PSNRs on clean data. However, VTM is not as effective as FR NIC in terms of PSNR with respect to $c(\mathcal{X})$ on this shift (the VTM curves start to flatten around 40-42 PSNR, while the NIC curves flatten around 42-47 PSNR). Like the snow corruption, all three models are comparably robust to the glass blur corruption (bottom row of Figure 12) and the severity of the corruption is the largest determinant of the PSNR with respect to $\mathcal{X}$.

On the high-frequency shift (shot noise), VTM displays a drastically different pattern from both NIC and JPEG2000. Specifically, VTM is able to achieve significantly higher PSNRs with respect to $c(\mathcal{X})$ (*i.e.*, generalize better) than FR NIC and JPEG2000. However, this comes at a very serious cost in

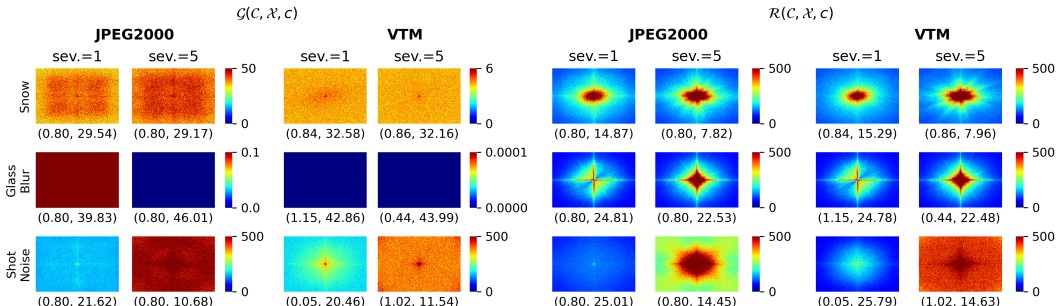

Figure 13: **Generalization error $\mathcal{G}$ and denoising error $\mathcal{R}$ for JPEG2000 and VTM.** We plot both spectral metrics for one low, medium, and high-frequency corruption at severities 1 and 5. Each plot is labeled with a tuple of that model's (bpp, PSNR) on the Kodak-C dataset with that corruption.

terms of bpp (notice the bpp range for shot noise in Figure 12 is 0-17 versus a typical range of 0-2). Recall that in Section 4.2.1 we observed that NIC models adjusted their bpp usage depending on the type of corruption– in particular, NIC models used more bpps than JPEG2000 on the shot noise corruption, even though it did not improve the PSNR. Figure 12 shows that VTM adjusts its bpp usage in a more extreme way than NIC models, but, unlike NIC models, this increase in bpp does result in an increase in PSNR with respect to $c(\mathcal{X})$. As expected from our results in Section 4.2.1, this increase in PSNR with respect to $c(\mathcal{X})$ results in a decrease of PSNR with respect $\mathcal{X}$. Overall, VTM's highly variable bpp usage could be a real disadvantage to using VTM in the wild: in particular, if a practitioner selects a hyper-parameter from clean data and expects the model will use a certain amount of bpps, VTM might use a much larger bpp than expected on OOD data. Thus, our OOD benchmark dataset is a vital aspect to comprehensive image compression testing.

Figure 13 shows the $\mathcal{G}$ and $\mathcal{R}$ metrics on the three representative shifts.[7] On the snow corruption, VTM has a smaller magnitude of errors than JPEG2000 across all frequencies; however, both models leave relatively even errors across all frequencies and both models distort low frequencies more than high frequencies (which is the opposite as FR NIC on the snow corruption in Figure 4). At both severities, the two models have similar plots of $\mathcal{G}$. VTM is highly effective at reconstructing the glass blur– the magnitude of $\mathcal{G}$ is <0.0001 across all frequencies– which matches our findings on the other codecs in Section 4.2.2. Additionally, JPEG2000 and VTM have nearly identical plots of $\mathcal{R}$ at both severities. Finally, on the shot noise corruption, both JPEG2000 and VTM display similar magnitudes and patterns of errors. Both models leave significantly more error on $\mathcal{G}$ at severity=5 than severity=1; however, JPEG2000 maintains a bpp=0.8 at both severities while VTM increases its bpp from 0.05 to 1.02. Additionally, unlike NIC, both of these models leave their highest errors in the low-frequency domain of $\mathcal{G}$. Lastly, both models are not as effective at denoising high-frequency signals as NIC proved to be.

# E    Additional results: ELIC

_____________________

[7]Note that we select one hyper-parameter for JPEG2000 (q=15); however, we vary the hyper-parameter choice for VTM across corruptions (q=32, 17, 47 for snow, glass blur, and shot noise respectively) so that VTM gives bpps in the ballpark of 0.8 for each corruption.

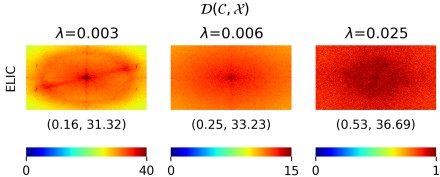

Figure 14: More detailed plots of $\mathcal{D}$ for ELIC. Equivalent to the ELIC row on the right side of Figure 2 with smaller colorbar ranges.

# F   Additional results: accuracies by individual corruptions

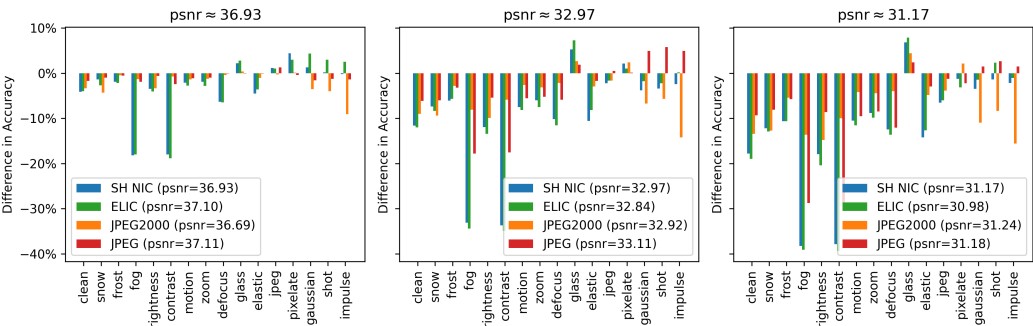

Figure 15: Effect of compressing corrupt images on classification accuracy. Specifically, let $A(X)$ be the top-1 accuracy of the model on dataset $X$, measured in percentage points. We report $A(\mathcal{C}(c(X))) - A(c(X))$ over all -C corruptions with severity 3 (or on clean ImageNet in the case of "clean"). Each subfigure shows results under a different fixed-PSNR constraint based on the PSNRs achieved by the compressors on the clean ImageNet dataset. Appendix M has tables with these accuracies.

# G    Additional results: Kodak and Kodak-C

This section contains Figures equivalent to those in the main body for the Kodak and Kodak-C dataset.

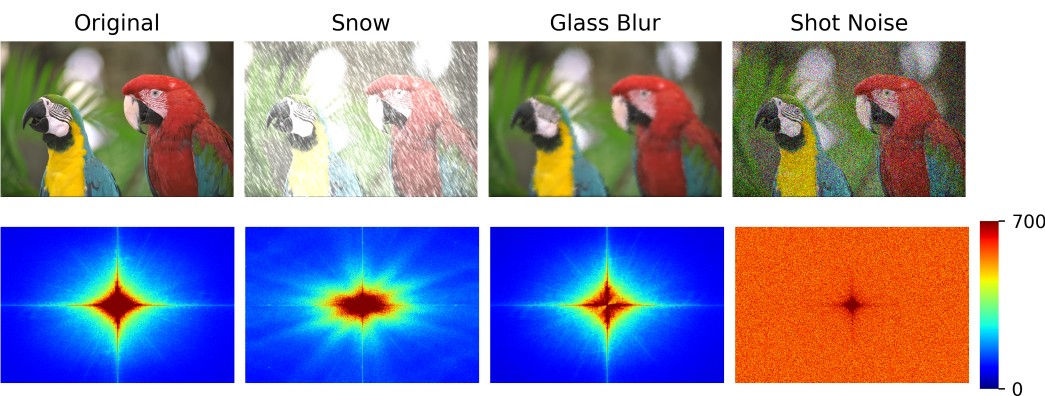

Figure 16: Figure equivalent to Figure 1a for Kodak-C dataset.

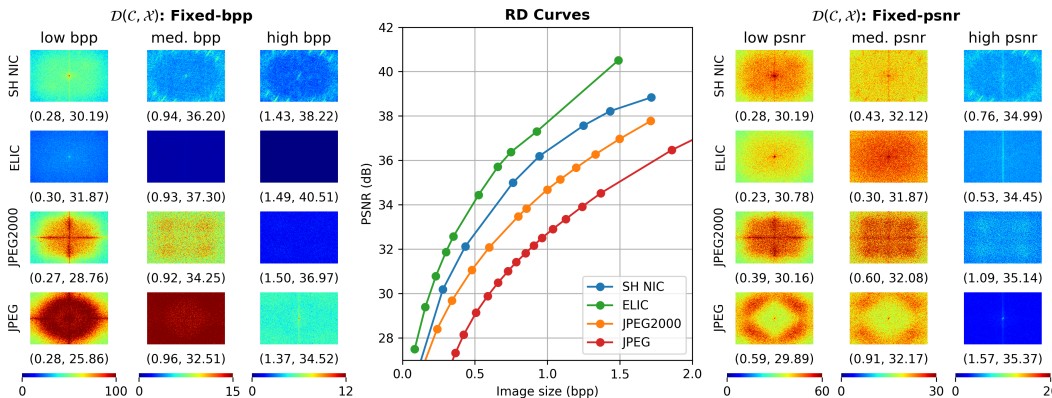

Figure 17: Figure equivalent to Figure 2 for Kodak dataset

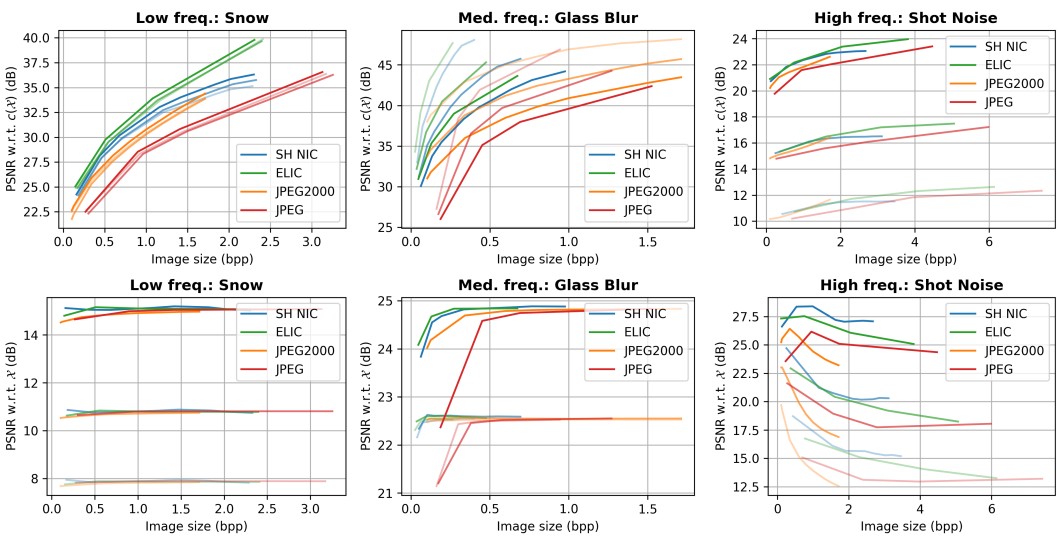

Figure 18: Figure equivalent to Figure 3 for Kodak-C dataset

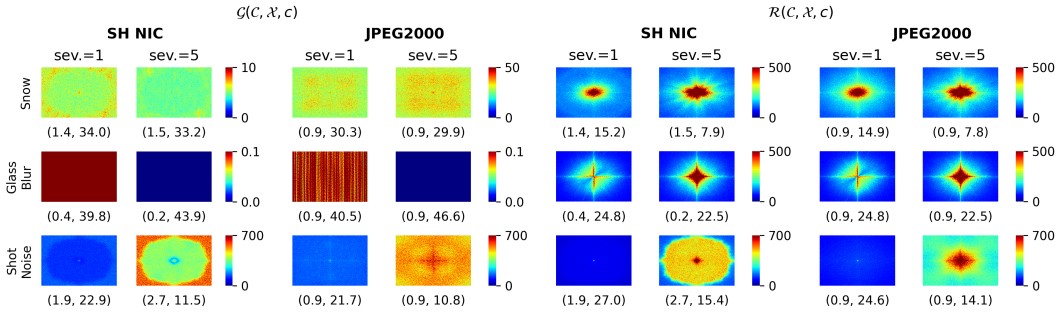

Figure 19: Figure equivalent to Figure 4 for Kodak-C dataset

# H  Additional results: other CLIC-C corruptions

Figures 20, 21, and 22 show the results of the other 12 corruptions from CLIC-C which were not included in the main body. Overall, we see that our three representative corruptions (snow, glass blur, and shot noise) represent the trends of the low, medium and high frequency corruptions well.

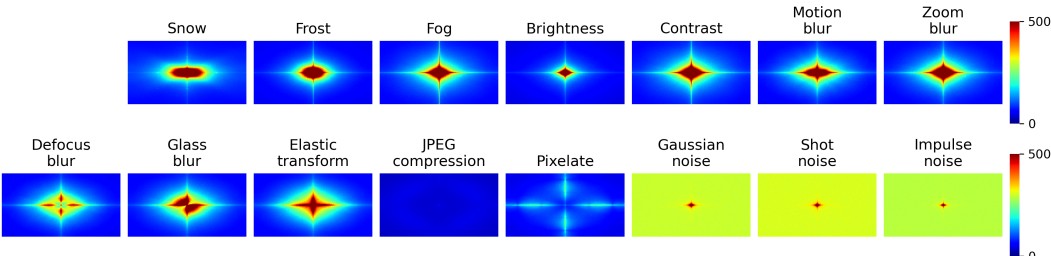

Figure 20: Average PSD of the difference between the corrupted images and the clean images for each given CLIC-C corruption $c$, $\frac{1}{N} \sum_{k=1}^{N} PSD(c(X_k) - X_k)$. Severity=5.

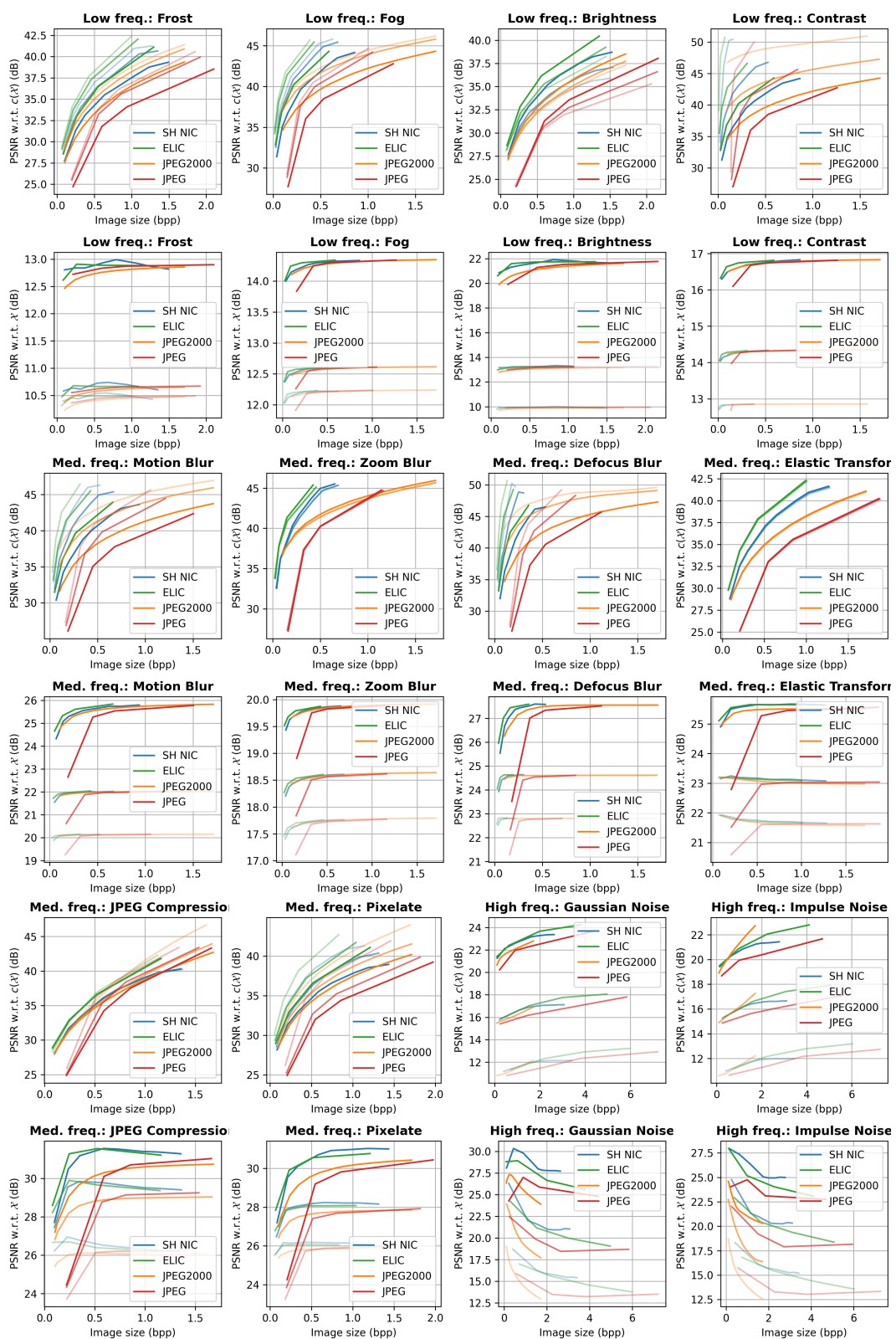

Figure 21: RD Curves for corruptions not seen in the main body. Rows 1, 3, and 5 show PSNR w.r.t. $c(\mathcal{X})$. Rows 2, 4, and 6 show PSNR w.r.t. $\mathcal{X}$.

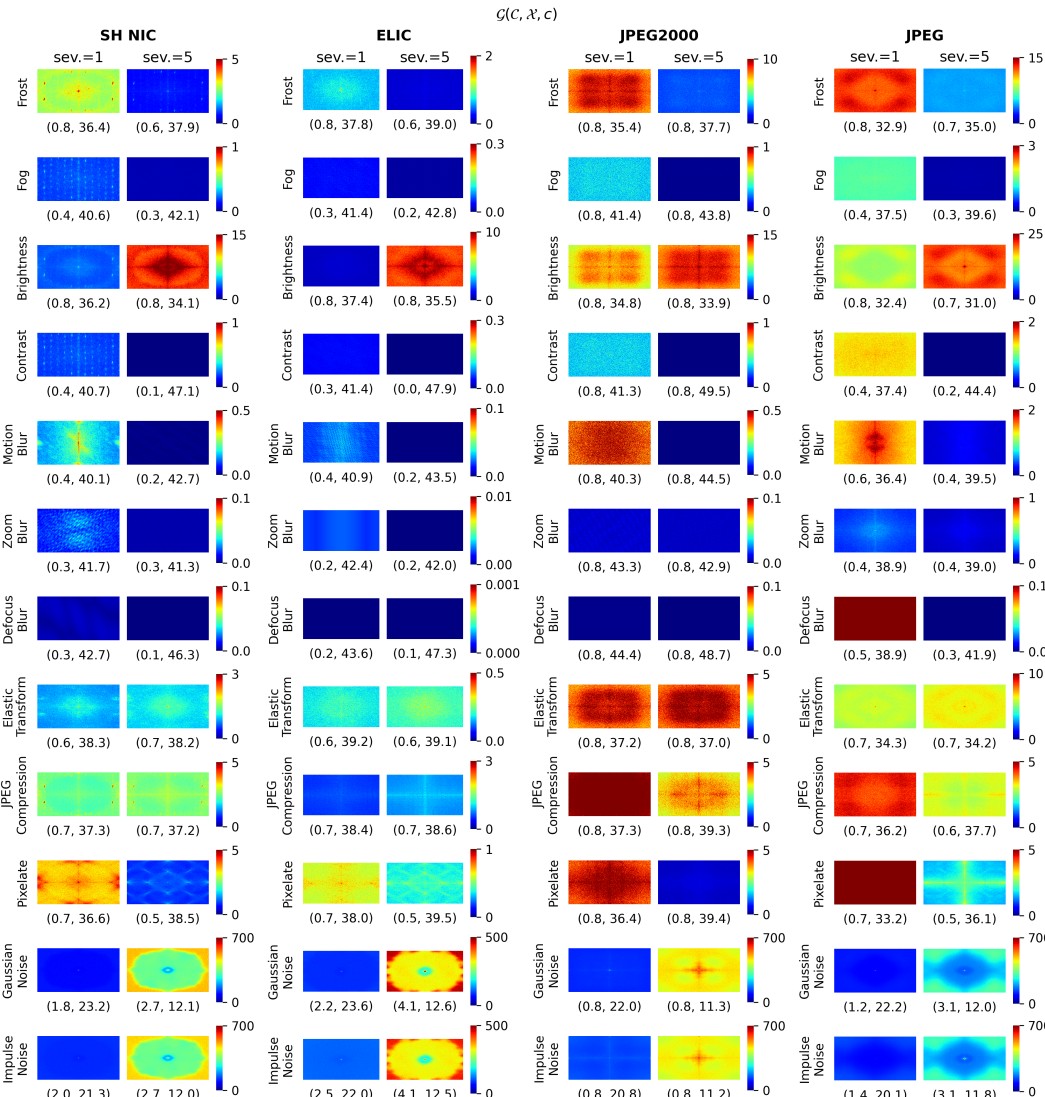

Figure 22: **Generalization error $\mathcal{G}$ on the other corruptions of CLIC-C.** We plot $\mathcal{G}$ for each corruption $c$ at severities 1 and 5. The plots of $\mathcal{G}$ are labeled with tuples of the model's (bpp, PSNR w.r.t. $c(\mathcal{X})$).

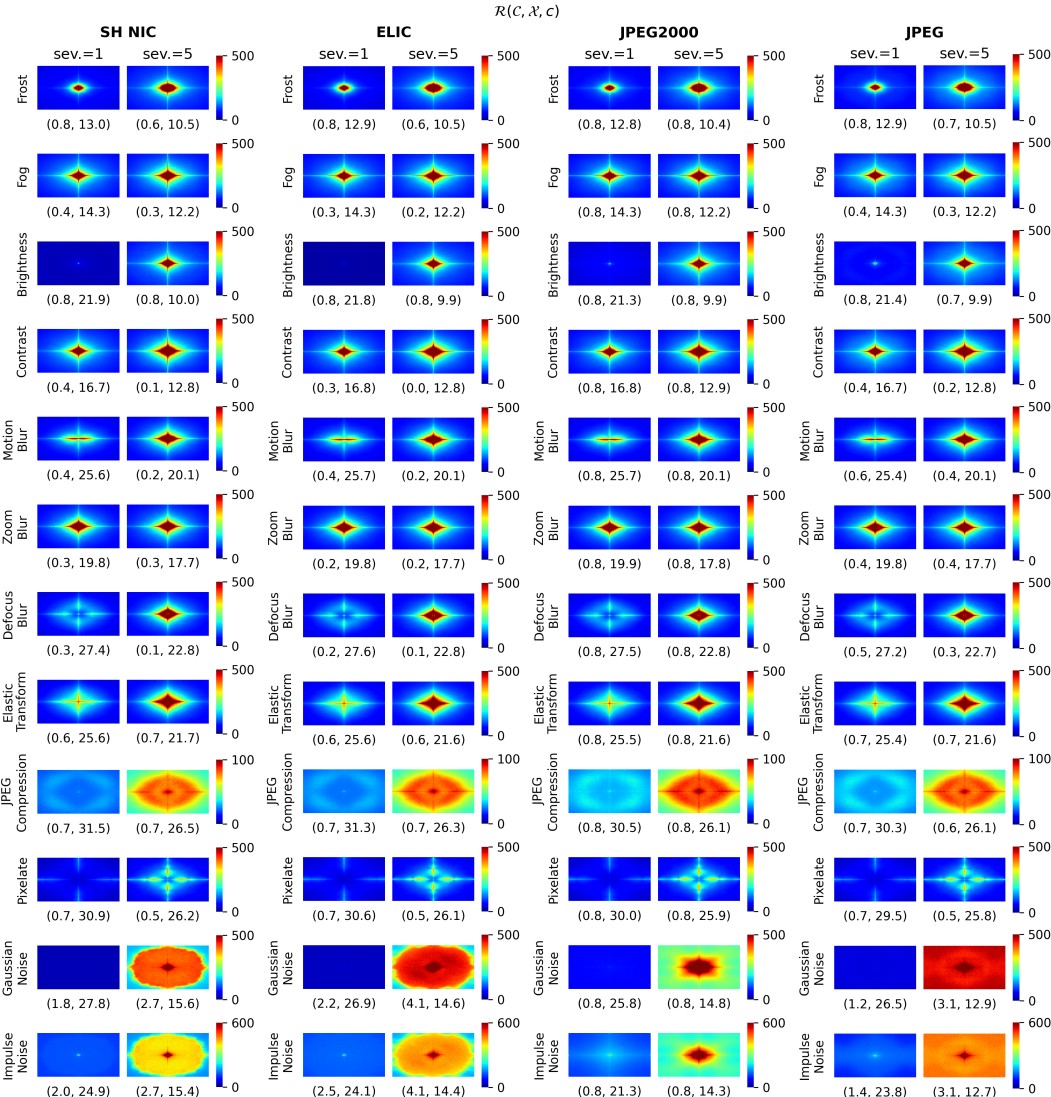

Figure 23: **Robustness error** $\mathcal{R}$ **on the other corruptions of CLIC-C.** We plot $\mathcal{R}$ for each corruption $c$ at severities 1 and 5. The plots of $\mathcal{G}$ are labeled with tuples of the model's (bpp, PSNR w.r.t. $\mathcal{X}$).

# I  Training setup and hyperparameters

We train NIC models using the train split of the CLIC 2020 dataset with batches of size 8 and random crops of size $256 \times 256$. For SH NIC, we set $N = M = 192$ and train for 5,000 epochs (about 1M iterations). We trained 8 models with $\lambda$s 0.0012, 0.005, 0.01, 0.03, 0.05, 0.1, 0.15, and 0.26. We used the pytorch model architecture in the compressai repository [10]. For ELIC, we use $N = 192, M = 320$ and train for 3,900 epochs. We trained 11 models with unique $\lambda$s (0.001, 0.0025, 0.003, 0.004, 0.006, 0.008, 0.016, 0.025, 0.032, 0.05, 0.15). We utilized this repository for the model architecture, but trained our own models on CLIC rather than experimenting with the publicly available checkpoints trained on ImageNet. Note that we trained more models for ELIC than FR NIC in order to obtain bpps and PSNRs which fit our fixed-bpp/PSNR constraint.

For the variable-rate models, we augment each convolutional layer with a FiLM layer which consists of two fully-connected layers and 128 hidden features. We train VR models for 10,000 epochs (about 2M iterations) and sample $\lambda$ from a log-uniform distribution over [0.0012, 0.26]. For pruned models, we use global gradual magnitude pruning (GMP) between epochs 675 and 2400 and tune the remaining weights from epochs 2401-10,000. Fixed-rate MS-SSIM models are trained over several values of $\lambda \in [1, 1000]$.

# J  Variable-rate model details

For our variable-rate NIC we implemented a model proposed by Dosovitskiy and Djolonga [17] which uses Loss Conditional Training (LCT) to train a variable-rate version of the scale-hyperprior NIC model from [9]. The model is trained over a continuous range of compression rates, optimizing the rate-distortion tradeoff at each. This reduces the computational redundancy involved in training separate models for the same task at different points along the rate-distortion curve. Additionally, once trained, this model can be adaptively used at any compression rate in the range, which makes it even more flexible than a discrete set of fixed-rate models.

In order to condition the compression rate via LCT, all convolutional layers in the model are augmented with Feature-wise Linear Modulation (FiLM) layers [49]. In this case, these are small neural networks that take a conditioning parameter $\lambda$ as input and output a $\boldsymbol{\mu}$ and $\boldsymbol{\sigma}$ used to modulate the activations channel-wise based on the value of $\lambda$. More specifically, suppose a layer in the CNN has activations $\boldsymbol{f}$ of size $W \times H \times C$. In Loss Conditional Training (LCT), these activations are augmented by $\boldsymbol{\mu}$ and $\boldsymbol{\sigma}$ as follows,

$$\tilde{\boldsymbol{f}} = \boldsymbol{\sigma}\mathbf{f} + \boldsymbol{\mu} \tag{1}$$

where both $\boldsymbol{\mu}$ and $\boldsymbol{\sigma}$ are vectors of size $C$.

Figure 24 shows a diagram of FiLM layers being applied to a small example network. Normally, without FiLM layers, the equation to calculate the features maps is

$$\mathbf{x}^{(1)} = \rho(W^{(1)}\mathbf{x}^{(0)} + \mathbf{b}^{(1)}), \tag{2}$$

where $\rho$ is the activation function (*e.g.*, ReLU), $W$ are the weights and $b$ are the biases in the layer. However, with FiLM layers, the weighted sum is first subject to augmentation with $\mu$ and $\sigma$. Thus, the new calculation is

$$\mathbf{x}^{(1)} = \rho(\boldsymbol{\sigma}^{(1)}(W^{(1)}\mathbf{x}^{(0)} + \mathbf{b}^{(1)})) + \boldsymbol{\mu}^{(1)}). \tag{3}$$

During training, a random $\lambda$ is drawn with each mini-batch and used in two ways: 1) as input to the FiLM layers to modulate the feature maps and 2) in the mini-batch's loss function. During inference, the model takes as input a desired compression rate $\lambda$ along with the input image. Notably, LCT and FiLM layers only affect the values of the activations (*i.e.*, feature maps), but not the *weights* on the convolutional layers or activation functions. This is an essential observation for applying pruning to a network with LCT.

# K  Pruned model details

One limitation of using NIC in the wild is the size of the NIC models. These models require several million learned weights and efficient inference requires specialized GPUs. This limits the possibility

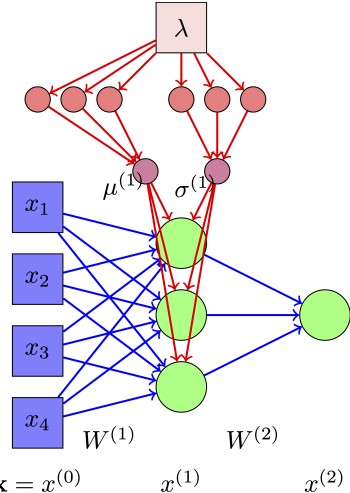

Figure 24: Cartoon of FiLM applied to a multi-layer perceptron. We consider pruning both the base layer weights (blue) and the film layer weights (red).

of using NIC on an edge device, like a drone or a cell phone. In this section, we demonstrate that NIC models themselves can be approximated (*i.e.*, we can compress the compressor) and we propose a sparse versions of both fixed-rate and variable-rate NIC models.

To prune our NIC models, we propose gradual magnitude pruning (GMP) [70] based approach which has been demonstrated to effectively sparsify DNNs. At a high level, GMP gradually prunes DNN weights based on their magnitude over a fixed number of epochs then continues to train, or fine-tune, the remaining sparse network after the target sparsity is reached. The pruning can be done layer-wise (*i.e.*, the mask must be updated such that each layer has the target sparsity) or globally (*i.e.*, the overall network must have the target sparsity, but the sparsities can differ between layers). We utilized global pruning, which has been demonstrated to produce more performant sparse models.

To implement GMP, each layer is given binary mask of the same size as the original layer, which specifies which weights have been pruned (*i.e.*, set to 0). Every $\Delta t$ iterations, the mask is updated based on the magnitudes of the weights, such that the network has a desired sparsity. Suppose $s_i$ is the initial sparsity (usually 0) at time $t_0$ and $s_t$ is the target sparsity (0.8 or 0.95 in our case) which we want to reach after $n$ steps, then the target sparsity at time step $t$ is

$$s_t = s_f + (s_i - s_f)\left(1 - \frac{t - t_0}{n\Delta t}\right)^3 \tag{4}$$

$$\text{for } t \in \{t_0, t_0 + \Delta t, \ldots, t_0 + n\Delta t\}. \tag{5}$$

The target sparsity equation is found to be especially effective because it causes weights to be pruned more quickly while the learning rate is still high and the weight-training on non-pruned weights can recover. After the desired sparsity is reached, the sparse network is trained for several more iterations. The pruning can be done layer-wise (*i.e.*, the mask must be updated such that each layer has the target sparsity) or globally (*i.e.*, the overall network must have the target sparsity, but the sparsities can differ between layers). Note that the network still performs weight training on the non-pruned weights, as usual, using an optimizer like stochastic gradient descent.

## L    Mathematical details

### L.1    Derivation of lemma 5.1

Let $\mathcal{X} \in \mathbb{R}^{N \times p}$ be a dataset (whose datapoints are the rows) with mean 0 and covariance $\Sigma = \frac{1}{N}\mathcal{X}^T\mathcal{X}$. Consider the problem of linearly auto-encoding $\mathcal{X}$ with latent space $\mathbb{R}^r$, where $r \leq p$:

$$\min_{W_1 \in M(r,p), W_2 \in M(p,r)} \frac{1}{2}|\mathcal{X}W_1^T W_2^T - \mathcal{X}|^2. \tag{6}$$

It is a theorem of Eckart and Young [18] that for any solution $W_1, W_2$ of eq. (6) the product $\mathcal{C} = W_2 W_1$ is an orthogonal projection onto the subspace spanned by the top $r$ principal components of $\mathcal{X}$ (see also [7] and for more recent developments and additional references, [50, 35]). Informally, this theorem says "linear autoencoders perform PCA." Formally, if $\mathcal{X} = USV^T$ is an SVD and $V[:r]$ is the first $r$ rows of $V$, then $W_2 W_1 = V[:r]^T V[:r]$. [8] It follows that the reconstruction error $\mathcal{L}(X)$ on a new input $X \in \mathbb{R}^p$ admits a simple closed form: let $V_1, \ldots, V_p$ be the principal components of $X$ (rows of $V$ above), and write

$$X = \sum_i c_i V_i; \text{ then } \mathcal{L}(X) = \sum_{i>r} c_i^2. \tag{7}$$

Suppose now that $\mathcal{X}$ is a dataset of natural images. We will replace the single index $i$ for $X = (X_1, \ldots, X_p) \in \mathbb{R}^p$ with 3 indices $h, i, j$ for an image tensor $X = (X_{h,i,j}) \in \mathbb{R}^{KHW}$ (here $h$ is viewed as the channel index and $i, j$ as the spatial indices); we will also commit mild abuse of notation by letting the matrices $W_i$ act on flattened (vectorized) tensors. Let $F : \mathbb{R}^{KHW} \to \mathbb{R}^{KHW}$ denote the spatial discrete Fourier transform. We introduce the notation

$$\hat{\mathcal{X}} = \mathcal{X}F^T \text{ and } \hat{\mathcal{C}} = F\mathcal{C}F^T, \tag{8}$$

and we will frequently use the shorthand "hat" decoration "$\hat{\phantom{-}}$" to denote $F$ as well, for ease of notation. Just as $\mathcal{C}$ projects onto the subspace spanned by the top $r$ principal components of $\mathcal{X}$, $\hat{\mathcal{C}}$ projects onto the subspace spanned by the top $r$ principal components of $\hat{\mathcal{X}}$.

It has long been known that when transformed to (spatial) Fourier frequency space, there are reasonable (coarse) descriptions of the statistical distribution of natural images. Its variances in spatial frequencies $(i, j)$ can be modelled as $\frac{1}{|i|^\alpha + |j|^\beta}$ where $\alpha, \beta \approx 2$ and its covariances drop off rapidly away from the diagonal [12, 1, 36].[9] These two facts motivate the following assumptions:

(I) The principal components of $\hat{\mathcal{X}}$ are roughly aligned with the spatial Fourier frequency basis.
(II) The associated variances are monotonically decreasing with frequency magnitude (more specifically according to the power law $\frac{1}{|i|^\alpha + |j|^\beta}$).

Assuming $\alpha = \beta = 2$, the above would suggest that $\hat{\mathcal{C}}$ projects a Fourier transformed image $\hat{X} = FX$ onto Fourier basis vectors corresponding to spatial frequencies $(i, j)$ with $i^2 + j^2 \leq \frac{r}{\pi K}$. That is,

$$(\hat{\mathcal{C}}\hat{X})_{:,ij} \approx \begin{cases} \hat{X}_{:,ij} & \text{if } i^2 + j^2 \leq \frac{r}{\pi K} \\ 0 & \text{otherwise.} \end{cases} \tag{9}$$

where $\hat{X}_{:,ij}$ denotes the components of $\hat{X}$ corresponding to spatial frequency $(i, j)$ (this is a vector of dimension $K$, the number of channels in the image dataset). Together with the identity $F\mathcal{C}X = \hat{\mathcal{C}}\hat{X}$, eq. (9) yields lemma 5.1.

*Remark* L.1. Note that lemma 5.1 also suggests that such autoencoder compressors behave qualitatively similarly to classical codecs such as JPEG, which involves a discrete cosine transform (DCT) followed by removal of high-frequency components [53]. While design of hand-crafted codecs like JPEG was no doubt driven by empirical observations of the statistics of natural images, the NIC models in our experiments (and at least in a theoretical sense the autoencoder model appearing in lemma 5.1) are *machine* learned from natural image datasets.

Below, we use the above analysis to get some qualitative insights into the robustness and generalization metrics $\mathcal{R}$ and $\mathcal{G}$ introduced in section 3. In what follows let $c : \mathbb{R}^{KHW} \to \mathbb{R}^{KHW}$ be a corruption transformation.

## L.2 Derivations of corollaries 5.2 and 5.3

**Robustness**: here we are interested in $\text{PSD}(X - \mathcal{C}(c(X)))$, which is simply the coordinatewise absolute value of $\hat{X} - \widehat{\mathcal{C}(c(X))}$. Using the fact that in our simple model $\mathcal{C}$ is linear, we can expand like

$$\hat{X} - \widehat{\mathcal{C}(c(X))} = \hat{X} - \hat{\mathcal{C}}(\widehat{c(X)}) = \hat{X} - \hat{\mathcal{C}}(\hat{X}) + \hat{\mathcal{C}}(\widehat{c(X)}) - \hat{X} \tag{10}$$

---

[8]although it is *in general not the case that* $W_1 = V[:r], W_2 = V[:r]^T$, see [50, §G] for discussion.

[9]Here we assume that our Fourier transform $F$ has the property that frequency $(0, 0)$ corresponds to constant images.

so that for any frequency $(i, j)$

$$
\begin{aligned}
|(\hat{X} - \mathcal{C}(\widehat{c(X)})):_{,ij}|^2 &= |(\hat{X} - \hat{\mathcal{C}}(\hat{X})):_{,ij}|^2 + 2\overline{(\hat{X} - \hat{\mathcal{C}}(\hat{X})):_{,ij}}\hat{\mathcal{C}}(\widehat{c(X)} - \hat{X}):_{,ij} \\
&\quad + |\hat{\mathcal{C}}(\widehat{c(X)}) - \hat{X}):_{,ij}|^2 \\
&\approx \begin{cases} |(\widehat{c(X)}) - \hat{X}):_{,ij}|^2 & \text{when } i^2 + j^2 \leq \frac{r}{\pi C} \\ |\hat{X}:_{,ij}|^2. & \text{otherwise.} \end{cases}
\end{aligned}
\tag{11}
$$

Taking square roots, we see that

$$
\mathrm{PSD}(X - \mathcal{C}(c(X))):_{,ij} \approx \begin{cases} |(\widehat{c(X)}) - \hat{X}):_{,ij}| & \text{when } i^2 + j^2 \leq \frac{r}{\pi C} \\ |\hat{X}:_{,ij}| & \text{otherwise;} \end{cases}
\tag{12}
$$

averaging over the dataset $\mathcal{X}$ gives corollary 5.2. In particular, the cross term $2\overline{(\hat{X} - \hat{\mathcal{C}}(\hat{X})):_{,ij}}\hat{\mathcal{C}}(\widehat{c(X)} - \hat{X}):_{,ij}$ is identically 0. This suggests that *autoencoder compressors trained on natural images are less robust to corruptions with large amplitude in low frequencies* (in the sense that $|(\widehat{c(X)} - \hat{X}):_{,ij}|^2$ is large for small values of $ij$).

**Generalization**: here we are interested in $\mathrm{PSD}(c(X) - \mathcal{C}(c(X)))$, which is simply the coordinatewise absolute value of $\widehat{c(X)} - \mathcal{C}(\widehat{c(X)})$. Again using the fact that in our simple model $\mathcal{C}$ is linear, we can expand like

$$
\begin{aligned}
\widehat{c(X)} - \mathcal{C}(\widehat{c(X)}) &= \hat{X} + (\widehat{c(X)} - \hat{X}) - \hat{\mathcal{C}}(\hat{X} + (\widehat{c(X)} - \hat{X})) \\
&= (\hat{X} - \hat{\mathcal{C}}(\hat{X})) + ((\widehat{c(X)} - \hat{X}) - \hat{\mathcal{C}}(\widehat{c(X)} - \hat{X})).
\end{aligned}
\tag{13}
$$

From this we obtain

$$
\begin{aligned}
&|(\widehat{c(X)} - \mathcal{C}(\widehat{c(X)})):_{,ij}|^2 \\
&= |(\hat{X} - \hat{\mathcal{C}}(\hat{X}))_{:,ij}|^2 + 2\overline{(\hat{X} - \hat{\mathcal{C}}(\hat{X}))_{:,ij}}((\widehat{c(X)} - \hat{X}) - \hat{\mathcal{C}}(\widehat{c(X)} - \hat{X}))_{:,ij} \\
&\quad + |((\widehat{c(X)} - \hat{X}) - \hat{\mathcal{C}}(\widehat{c(X)} - \hat{X}))_{:,ij}|^2 \\
&\approx \begin{cases} 0 & \text{when } i^2 + j^2 \leq \frac{r}{\pi C} \\ |\hat{X}:_{,ij}|^2 + 2\overline{\hat{X}:_{,ij}}(\widehat{c(X)} - \hat{X}):_{,ij} + |(\widehat{c(X)} - \hat{X}):_{,ij}|^2 & \text{otherwise.} \end{cases}
\end{aligned}
\tag{14}
$$

Taking square roots and relating back to power spectral density, this says

$$
\begin{aligned}
&\mathrm{PSD}(c(X) - \mathcal{C}(c(X))):_{,ij} \\
&\approx \begin{cases} 0 & \text{when } i^2 + j^2 \leq \frac{r}{\pi C} \\ \sqrt{|\hat{X}:_{,ij}|^2 + 2\overline{\hat{X}:_{,ij}}(\widehat{c(X)} - \hat{X}):_{,ij} + |(\widehat{c(X)} - \hat{X}):_{,ij}|^2} & \text{otherwise} \end{cases}
\end{aligned}
\tag{15}
$$

and averaging over the dataset $\mathcal{X}$ gives corollary 5.3. Equation (14) is more involved than eq. (11) – in particular, the "cross term" $2\overline{\hat{X}:_{,ij}}(\widehat{c(X)} - \hat{X}):_{,ij}$ is in general non-zero. However, there are many cases where at least an expectation over the data set $\hat{\mathcal{X}}$ the term $2\overline{\hat{X}:_{,ij}}(\widehat{c(X)} - \hat{X}):_{,ij}$ vanishes (for example, when $c$ is additive noise, or more generally when $\widehat{c(X)} - \hat{X}$ is statistically independent of $\hat{X}$.). In such cases, we can see that *autoencoder compressors trained on natural images generalize less successfully to corruptions with large amplitude in high frequencies*.

### L.3 Autoencoder reconstruction error and data density more generally

Our findings on *generalization* (both the experimental results and analysis with linear autoencoders) is closely related to the widely observed *inverse correlation* between autoencoder reconstruction error and probability density. We will take the recent [29] as a jumping-off point. However, it is worth noting that this observation (or at least something similar to it) has been around for some time, at least since [5], and forms the basis for widespread use of autoencoders in anomaly detection.

Formally, [29, Observation 2] states that:

(∗) For an autoencoder $\mathcal{C}$ the reconstruction error $|\mathcal{C}(X) - X|_2$ is positively correlated with $\frac{1}{p(X)}$, the probability density of the input data at $X$.

If we assume this observation holds, then some of our experimental findings regarding generalization to corrupted data admit a simple explanation: as discussed above, the probability density of natural images is heavily concentrated along low (spatial) Fourier frequencies. Hence for a corruption $c$ such that $c(X) - X$ is concentrated in high frequencies, it is reasonable to suspect that $p(c(X)) < p(X)$, and given the assumed positive correlation of $\frac{1}{p(X)}$ with $|\mathcal{C}(X) - X|_2$, that

$$|\mathcal{C}(c(X)) - c(X)|_2 > |\mathcal{C}(X) - X|_2, \tag{16}$$

i.e. the reconstruction error of the corrupted datapoint, related to our generalization metric, is higher than that of the clean datapoint $X$.

*Remark* L.2. In some sense, (∗) is a simple consequence of learning via risk minimization. Let $\mathcal{C} : M \to M$ be an autoencoder on a manifold $M$ with finite *volume* (for example, the image hypercube $[0, 1]^{CHW}$). Let $\ell(\mathcal{C}(X), X)$ be a reconstruction loss on a point $X \in M$ and let $p(X)$ be the probability density of input data on $M$, assumed to be positive everywhere (otherwise, discussing correlation with $\frac{1}{p(X)}$ is troublesome). The risk in this situation is

$$\mathcal{L}(\mathcal{C}) := \int_M \ell(\mathcal{C}(X), X) p(X) dx \tag{17}$$

Note that $\ell(\mathcal{C}(X), X)$ and $\frac{1}{p(X)}$ will be positively correlated provided that their covariance is positive. By definition this covariance is

$$\int_M \ell(\mathcal{C}(X), X) \frac{1}{p(X)} p(X) dx - \int_M \ell(\mathcal{C}(X), X) p(X) dx \cdot \int_M \frac{1}{p(X)} p(X) dx$$
$$= \int_M \ell(\mathcal{C}(X), X) dx - \mathcal{L}(\mathcal{C}) \cdot \mathrm{vol}M. \tag{18}$$

Since at present we only care about the sign of the covariance, we can divide by $\mathrm{vol}M$ to obtain

$$\int_M \ell(\mathcal{C}(X), X) \frac{dx}{\mathrm{vol}M} - \mathcal{L}(\mathcal{C}); \tag{19}$$

the first term is the risk of $\mathcal{C}$ with respect to the *uniform* distribution on $M$. In words, eq. (19) is positive whenever $\mathcal{C}$ performs better on the distribution $p(X)$ than the uniform distribution, and this outcome is to be expected for an autoencoder with sufficient capacity trained on a non-uniform distribution $p(X)$.

*Remark* L.3. With notation as in the previous example, the result [4, Thm. 2] shows that the solution obtained with calculus of variations applied to an objective of the form

$$\min_{\mathcal{C}} \int_M \left( |\mathcal{C}(X) - X|_2^2 + \sigma^2 |\nabla_X \mathcal{C}(X)|_2^2 \right) p(X) dx \tag{20}$$

(i.e. mean squared error loss and an $\ell^2$ penalty on gradient norms) satisfies

$$\mathcal{C}(X) - X = \sigma^2 \nabla_X \log p(X) + o(\sigma^2) \text{ and} \tag{21}$$
$$\nabla_X \mathcal{C}(X) = I + \sigma^2 \mathrm{Hess}_X \log p(X) + o(\sigma^2) \text{ as } \sigma \to 0. \tag{22}$$

In particular, the reconstruction error $|\mathcal{C}(X) - X|_2$ is proportional to the gradient norm $|\nabla_X \log p(X)|_2$ (up to higher order terms in $\sigma^2$ as $\sigma \to 0$). Since

$$|\nabla_X \log p(X)|_2 = \frac{1}{p(X)} |\nabla_X p(X)|_2,$$

this result sheds a more precise and quantitative light on (∗).

Equation (22) may also shed some light on robustness of $\mathcal{C}$ as measured by $\mathcal{R}$: indeed, if we assume $c(X) - X$ is small,

$$\mathcal{C}(c(X)) - X = \mathcal{C}(X + c(X) - X) - X \approx \mathcal{C}(X) + \nabla_X \mathcal{C}^T (c(X) - X) - X. \tag{23}$$

Applying eqs. (21) and (22) gives the approximation

$$
\begin{aligned}
\mathcal{C}(X) + \nabla_X \mathcal{C}^T(c(X) - X) - X &\approx \sigma^2 \nabla_X \log p(X) \\
&\quad + (I + \sigma^2 \operatorname{Hess}_X \log p(X))(c(X) - X) + o(\sigma^2)
\end{aligned}
\tag{24}
$$

as $\sigma \to 0$. Taking norms squared we get

$$
\begin{aligned}
|\mathcal{C}(c(X)) - X|^2 &\approx \sigma^2 |\nabla_X \log p(X)|^2 \\
&\quad + 2\sigma^2 \nabla_X \log p(X)^T (I + \sigma^2 \operatorname{Hess}_X \log p(X))(c(X) - X) \\
&\quad + |(I + \sigma^2 \operatorname{Hess}_X \log p(X))(c(X) - X)|^2 + o(\sigma^2)
\end{aligned}
\tag{25}
$$

as $\sigma \to 0$. If we again make the simplifying assumption that $c(X) - X$ is random and independent of $X$, then after taking the expectation over $c$ (here denoted by $E_c$) we obtain

$$
\begin{aligned}
E_c[|\mathcal{C}(c(X)) - X|^2] &\approx \sigma^2 |\nabla_X \log p(X)|^2 \\
&\quad + E_c[|(I + \sigma^2 \operatorname{Hess}_X \log p(X))(c(X) - X)|^2] + o(\sigma^2)
\end{aligned}
\tag{26}
$$

as $\sigma \to 0$. If we let $\alpha_1 \geq \alpha_2 \geq \cdots \geq \alpha_p$ be the eigenvalues of $\operatorname{Hess}_X \log p(X)$ and choose $\sigma$ small enough that $1 + \frac{\sigma^2}{\alpha}_i > 0$ for all $i$, we see that the norm

$$
|(I + \sigma^2 \operatorname{Hess}_X \log p(X))(c(X) - X)|^2
\tag{27}
$$

*surpresses* (resp. *magnifies*) contributions from components of $c(X) - X$ along the eigenvectors associated with $\alpha_i$ for large (resp. small) $i$. Note that it is reasonable as a first approximation to expect $\operatorname{Hess}_X \log p(X)$ to be negative definite (this is precisely the case when $p$ is a Gaussian distribution, and more generally the requirements $p(X) \geq 0$ and $\int p(X)dX = 1$ put restrictions on the extent to which $\operatorname{Hess}_X \log p(X)$ can exhibit non-negative eigenvalues). In this case the large (resp. small) values of $i$ correspond to the eigenvectors in directions of "sharpest" (resp. "shallowest") curvature of the graph of $p(X)$. Explicitly, if $X$ is Gaussian with covariance $\Sigma$, then $Hess_X \log p(X) = \Sigma^{-1}$, and if $\lambda_1 \geq \lambda_2 \geq \cdots \geq \lambda_p$ are the eigenvalues of $\Sigma$ then $\alpha_i = -\frac{1}{\lambda_i}$ for all $i$. In this case we see that the norm in eq. (27) *surpresses* (resp. *magnifies*) contributions from components of $c(X) - X$ along principal components with low (resp. high) variance. Note that this provides a takeaway qualitatively similar to that of corollary 5.2.

## L.4 A fundamental robustness-generalization inequality

The similarity between the $\mathcal{R}$ heatmaps for glass blur in fig. 4 and the PSDs of the corruptions themselves appearing in fig. 1a has a simple explanation in terms of a fundamental relationship between $\mathcal{D}, \mathcal{G}$ and $\mathcal{R}$. One can show (via two applications of the triangle inequality) that

$$
\frac{1}{N} \sum_{k=1}^{N} PSD(X_k - c(X_k)) - \mathcal{G}(\mathcal{C}, \mathcal{X}, c) \leq \mathcal{R}(\mathcal{C}, \mathcal{X}, c) \leq \frac{1}{N} \sum_{k=1}^{N} PSD(X_k - c(X_k)) + \mathcal{G}(\mathcal{C}, \mathcal{X}, c).
\tag{28}
$$

In the case of glass blur, generalization error is very low, so eq. (28) reduces to $\mathcal{R} \approx \frac{1}{N} \sum_{k=1}^{N} PSD(X_k - c(X_k))$, where the right-hand side is the average PSD of the corruption itself.

## M Tables of classification accuracies

|                   | No Compression |
|-------------------|----------------|
| clean             | 0.76           |
| snow              | 0.34           |
| frost             | 0.31           |
| fog               | 0.46           |
| brightness        | 0.69           |
| contrast          | 0.44           |
| motion blur       | 0.37           |
| zoom blur         | 0.35           |
| defocus blur      | 0.36           |
| glass blur        | 0.16           |
| elastic transform | 0.53           |
| jpeg compression  | 0.59           |
| pixelate          | 0.50           |
| gaussian noise    | 0.32           |
| shot noise        | 0.29           |
| impulse noise     | 0.32           |

Table 1: Classification accuracies without compression on the validation split of ImageNet-C.

|                   | SH NIC | ELIC | JPEG2000 | JPEG |
|-------------------|--------|------|----------|------|
| **PSNR=36.93**    |        |      |          |      |
| clean             | 0.72   | 0.72 | 0.72     | 0.74 |
| snow              | 0.33   | 0.31 | 0.30     | 0.33 |
| frost             | 0.29   | 0.29 | 0.30     | 0.30 |
| fog               | 0.28   | 0.28 | 0.45     | 0.44 |
| brightness        | 0.65   | 0.65 | 0.65     | 0.68 |
| contrast          | 0.26   | 0.26 | 0.44     | 0.42 |
| motion blur       | 0.35   | 0.34 | 0.36     | 0.36 |
| zoom blur         | 0.33   | 0.32 | 0.34     | 0.34 |
| defocus blur      | 0.30   | 0.30 | 0.36     | 0.36 |
| glass blur        | 0.19   | 0.19 | 0.17     | 0.16 |
| elastic transform | 0.48   | 0.49 | 0.52     | 0.53 |
| jpeg compression  | 0.60   | 0.60 | 0.59     | 0.60 |
| pixelate          | 0.55   | 0.53 | 0.51     | 0.50 |
| gaussian noise    | 0.33   | 0.36 | 0.28     | 0.30 |
| shot noise        | 0.29   | 0.32 | 0.25     | 0.28 |
| impulse noise     | 0.31   | 0.34 | 0.23     | 0.30 |
| **PSNR=32.97**    |        |      |          |      |
| clean             | 0.64   | 0.64 | 0.67     | 0.70 |
| snow              | 0.27   | 0.26 | 0.25     | 0.28 |
| frost             | 0.25   | 0.25 | 0.28     | 0.28 |
| fog               | 0.13   | 0.11 | 0.38     | 0.28 |
| brightness        | 0.57   | 0.55 | 0.59     | 0.63 |
| contrast          | 0.11   | 0.09 | 0.38     | 0.27 |
| motion blur       | 0.30   | 0.29 | 0.34     | 0.32 |
| zoom blur         | 0.29   | 0.28 | 0.32     | 0.30 |
| defocus blur      | 0.26   | 0.24 | 0.34     | 0.30 |
| glass blur        | 0.22   | 0.24 | 0.19     | 0.18 |
| elastic transform | 0.42   | 0.45 | 0.50     | 0.51 |
| jpeg compression  | 0.57   | 0.57 | 0.57     | 0.59 |
| pixelate          | 0.53   | 0.51 | 0.53     | 0.51 |
| gaussian noise    | 0.28   | 0.30 | 0.25     | 0.37 |
| shot noise        | 0.26   | 0.27 | 0.23     | 0.35 |
| impulse noise     | 0.29   | 0.32 | 0.17     | 0.37 |
| **PSNR=31.17**    |        |      |          |      |
| clean             | 0.58   | 0.57 | 0.62     | 0.66 |
| snow              | 0.22   | 0.21 | 0.21     | 0.26 |
| frost             | 0.20   | 0.20 | 0.25     | 0.25 |
| fog               | 0.08   | 0.07 | 0.32     | 0.17 |
| brightness        | 0.51   | 0.48 | 0.54     | 0.60 |
| contrast          | 0.07   | 0.05 | 0.34     | 0.16 |
| motion blur       | 0.27   | 0.26 | 0.33     | 0.28 |
| zoom blur         | 0.26   | 0.25 | 0.31     | 0.27 |
| defocus blur      | 0.24   | 0.22 | 0.32     | 0.24 |
| glass blur        | 0.23   | 0.24 | 0.21     | 0.19 |
| elastic transform | 0.39   | 0.40 | 0.48     | 0.50 |
| jpeg compression  | 0.52   | 0.53 | 0.55     | 0.58 |
| pixelate          | 0.49   | 0.47 | 0.53     | 0.48 |
| gaussian noise    | 0.28   | 0.30 | 0.21     | 0.33 |
| shot noise        | 0.28   | 0.31 | 0.21     | 0.32 |
| impulse noise     | 0.29   | 0.31 | 0.16     | 0.33 |

Table 2: Classification accuracies from Figures 5 and 15

| | FR NIC | VR NIC | VR NIC pr=0.8 | VR NIC pr=0.95 | FR NIC opt. MS-SSIM | JPEG2000 |
|---|---|---|---|---|---|---|
| **bpp = 1.23** | | | | | | |
| clean | 0.74 | 0.73 | 0.73 | 0.73 | 0.74 | 0.72 |
| snow | 0.33 | 0.33 | 0.33 | 0.32 | 0.32 | 0.30 |
| frost | 0.30 | 0.30 | 0.29 | 0.28 | 0.30 | 0.30 |
| fog | 0.36 | 0.36 | 0.36 | 0.36 | 0.41 | 0.45 |
| brightness | 0.67 | 0.67 | 0.66 | 0.66 | 0.67 | 0.65 |
| contrast | 0.36 | 0.35 | 0.34 | 0.36 | 0.40 | 0.44 |
| motion blur | 0.36 | 0.36 | 0.35 | 0.36 | 0.37 | 0.36 |
| zoom blur | 0.34 | 0.34 | 0.33 | 0.34 | 0.35 | 0.34 |
| defocus blur | 0.32 | 0.32 | 0.32 | 0.33 | 0.34 | 0.36 |
| glass blur | 0.17 | 0.17 | 0.16 | 0.16 | 0.16 | 0.17 |
| elastic transform | 0.50 | 0.50 | 0.49 | 0.49 | 0.50 | 0.52 |
| jpeg compression | 0.60 | 0.60 | 0.61 | 0.61 | 0.60 | 0.59 |
| pixelate | 0.53 | 0.53 | 0.54 | 0.54 | 0.58 | 0.51 |
| gaussian noise | 0.34 | 0.33 | 0.33 | 0.35 | 0.36 | 0.28 |
| shot noise | 0.31 | 0.30 | 0.31 | 0.33 | 0.32 | 0.25 |
| impulse noise | 0.32 | 0.31 | 0.32 | 0.34 | 0.35 | 0.23 |
| **bpp = 0.77** | | | | | | |
| clean | 0.72 | 0.71 | 0.71 | 0.71 | 0.71 | 0.69 |
| snow | 0.33 | 0.32 | 0.32 | 0.32 | 0.29 | 0.27 |
| frost | 0.29 | 0.29 | 0.28 | 0.28 | 0.27 | 0.29 |
| fog | 0.28 | 0.27 | 0.27 | 0.25 | 0.36 | 0.41 |
| brightness | 0.65 | 0.65 | 0.65 | 0.64 | 0.65 | 0.62 |
| contrast | 0.26 | 0.25 | 0.26 | 0.24 | 0.37 | 0.41 |
| motion blur | 0.35 | 0.35 | 0.34 | 0.34 | 0.36 | 0.35 |
| zoom blur | 0.33 | 0.33 | 0.32 | 0.32 | 0.35 | 0.33 |
| defocus blur | 0.30 | 0.29 | 0.29 | 0.29 | 0.33 | 0.35 |
| glass blur | 0.19 | 0.19 | 0.17 | 0.17 | 0.17 | 0.18 |
| elastic transform | 0.48 | 0.48 | 0.48 | 0.47 | 0.48 | 0.51 |
| jpeg compression | 0.60 | 0.60 | 0.60 | 0.60 | 0.59 | 0.58 |
| pixelate | 0.55 | 0.55 | 0.54 | 0.54 | 0.60 | 0.52 |
| gaussian noise | 0.33 | 0.32 | 0.32 | 0.34 | 0.34 | 0.31 |
| shot noise | 0.29 | 0.29 | 0.29 | 0.32 | 0.31 | 0.28 |
| impulse noise | 0.31 | 0.30 | 0.31 | 0.33 | 0.33 | 0.26 |
| **bpp = 0.24** | | | | | | |
| clean | 0.58 | 0.56 | 0.58 | 0.56 | 0.60 | 0.52 |
| snow | 0.22 | 0.22 | 0.22 | 0.21 | 0.18 | 0.14 |
| frost | 0.20 | 0.20 | 0.20 | 0.20 | 0.20 | 0.19 |
| fog | 0.08 | 0.06 | 0.06 | 0.05 | 0.18 | 0.21 |
| brightness | 0.51 | 0.49 | 0.50 | 0.49 | 0.53 | 0.43 |
| contrast | 0.07 | 0.05 | 0.05 | 0.05 | 0.20 | 0.25 |
| motion blur | 0.27 | 0.25 | 0.26 | 0.25 | 0.30 | 0.29 |
| zoom blur | 0.26 | 0.25 | 0.26 | 0.25 | 0.29 | 0.27 |
| defocus blur | 0.24 | 0.22 | 0.22 | 0.21 | 0.27 | 0.28 |
| glass blur | 0.23 | 0.21 | 0.19 | 0.19 | 0.24 | 0.24 |
| elastic transform | 0.39 | 0.37 | 0.37 | 0.36 | 0.42 | 0.42 |
| jpeg compression | 0.52 | 0.51 | 0.52 | 0.51 | 0.53 | 0.47 |
| pixelate | 0.49 | 0.47 | 0.48 | 0.47 | 0.54 | 0.47 |
| gaussian noise | 0.28 | 0.28 | 0.25 | 0.25 | 0.27 | 0.27 |
| shot noise | 0.28 | 0.27 | 0.25 | 0.25 | 0.28 | 0.26 |
| impulse noise | 0.29 | 0.28 | 0.25 | 0.25 | 0.26 | 0.26 |

Table 3: Classification accuracies from Figure 9

| | FR NIC | VR NIC | VR NIC pr=0.8 | VR NIC pr=0.95 | FR NIC opt. MS-SSIM | JPEG2000 |
|---|---|---|---|---|---|---|
| **PSNR=36.93** | | | | | | |
| clean | 0.72 | 0.71 | 0.73 | 0.73 | 0.74 | 0.72 |
| snow | 0.33 | 0.32 | 0.33 | 0.32 | 0.33 | 0.30 |
| frost | 0.29 | 0.29 | 0.29 | 0.28 | 0.30 | 0.30 |
| fog | 0.28 | 0.27 | 0.33 | 0.35 | 0.42 | 0.45 |
| brightness | 0.65 | 0.65 | 0.66 | 0.66 | 0.67 | 0.65 |
| contrast | 0.26 | 0.25 | 0.31 | 0.34 | 0.41 | 0.44 |
| motion blur | 0.35 | 0.35 | 0.35 | 0.36 | 0.37 | 0.36 |
| zoom blur | 0.33 | 0.33 | 0.33 | 0.34 | 0.35 | 0.34 |
| defocus blur | 0.30 | 0.29 | 0.30 | 0.32 | 0.35 | 0.36 |
| glass blur | 0.19 | 0.19 | 0.16 | 0.16 | 0.16 | 0.17 |
| elastic transform | 0.48 | 0.48 | 0.49 | 0.49 | 0.50 | 0.52 |
| jpeg compression | 0.60 | 0.60 | 0.61 | 0.61 | 0.60 | 0.59 |
| pixelate | 0.55 | 0.55 | 0.54 | 0.54 | 0.56 | 0.51 |
| gaussian noise | 0.33 | 0.32 | 0.33 | 0.35 | 0.34 | 0.28 |
| shot noise | 0.29 | 0.29 | 0.30 | 0.33 | 0.31 | 0.25 |
| impulse noise | 0.31 | 0.30 | 0.31 | 0.34 | 0.33 | 0.23 |
| **PSNR=32.97** | | | | | | |
| clean | 0.64 | 0.64 | 0.66 | 0.66 | 0.71 | 0.67 |
| snow | 0.27 | 0.27 | 0.28 | 0.29 | 0.28 | 0.25 |
| frost | 0.25 | 0.25 | 0.26 | 0.26 | 0.27 | 0.28 |
| fog | 0.13 | 0.12 | 0.13 | 0.14 | 0.35 | 0.38 |
| brightness | 0.57 | 0.57 | 0.59 | 0.60 | 0.64 | 0.59 |
| contrast | 0.11 | 0.10 | 0.12 | 0.13 | 0.36 | 0.38 |
| motion blur | 0.30 | 0.30 | 0.30 | 0.31 | 0.36 | 0.34 |
| zoom blur | 0.29 | 0.29 | 0.29 | 0.29 | 0.35 | 0.32 |
| defocus blur | 0.26 | 0.25 | 0.24 | 0.25 | 0.32 | 0.34 |
| glass blur | 0.22 | 0.21 | 0.18 | 0.19 | 0.18 | 0.19 |
| elastic transform | 0.42 | 0.43 | 0.43 | 0.43 | 0.48 | 0.50 |
| jpeg compression | 0.57 | 0.57 | 0.58 | 0.58 | 0.59 | 0.57 |
| pixelate | 0.53 | 0.52 | 0.53 | 0.53 | 0.60 | 0.53 |
| gaussian noise | 0.28 | 0.29 | 0.28 | 0.31 | 0.33 | 0.25 |
| shot noise | 0.26 | 0.26 | 0.26 | 0.28 | 0.31 | 0.23 |
| impulse noise | 0.29 | 0.29 | 0.28 | 0.30 | 0.33 | 0.17 |
| **PSNR=31.17** | | | | | | |
| clean | 0.58 | 0.58 | 0.61 | 0.62 | 0.67 | 0.62 |
| snow | 0.22 | 0.23 | 0.24 | 0.25 | 0.25 | 0.21 |
| frost | 0.20 | 0.22 | 0.22 | 0.23 | 0.25 | 0.25 |
| fog | 0.08 | 0.07 | 0.08 | 0.08 | 0.29 | 0.32 |
| brightness | 0.51 | 0.51 | 0.54 | 0.55 | 0.60 | 0.54 |
| contrast | 0.07 | 0.06 | 0.07 | 0.08 | 0.30 | 0.34 |
| motion blur | 0.27 | 0.26 | 0.28 | 0.28 | 0.34 | 0.33 |
| zoom blur | 0.26 | 0.26 | 0.27 | 0.27 | 0.33 | 0.31 |
| defocus blur | 0.24 | 0.23 | 0.23 | 0.23 | 0.30 | 0.32 |
| glass blur | 0.23 | 0.21 | 0.19 | 0.19 | 0.20 | 0.21 |
| elastic transform | 0.39 | 0.39 | 0.39 | 0.40 | 0.46 | 0.48 |
| jpeg compression | 0.52 | 0.53 | 0.55 | 0.55 | 0.57 | 0.55 |
| pixelate | 0.49 | 0.48 | 0.50 | 0.50 | 0.59 | 0.53 |
| gaussian noise | 0.28 | 0.28 | 0.25 | 0.26 | 0.30 | 0.21 |
| shot noise | 0.28 | 0.26 | 0.25 | 0.25 | 0.30 | 0.21 |
| impulse noise | 0.29 | 0.28 | 0.26 | 0.27 | 0.30 | 0.16 |

Table 4: Classification accuracies from Figure 9

