# OpenReview forum: "Neural Image Compression: Generalization, Robustness, and Spectral Biases"
_NeurIPS.cc/2023/Conference — NeurIPS 2023 poster_

### Official Review · Reviewer_fKAX · 2023-07-06

**Soundness:** 3 good
**Presentation:** 4 excellent
**Contribution:** 2 fair
**Rating:** 5
**Confidence:** 4

**Summary:**

This paper proposes to adopt spectral tools to analysis OOD issues of NIC. On the proposed corrupted dataset, the paper investigates the error of both conventional and learning-based image compression approaches and provides some interesting conclusions, showing robustness and generalization of NIC and suggesting insights of design a better NIC model in the wild.

**Strengths:**

The paper proposes novel tools and benchmarks to thoroughly study NIC from the OOD perspective. I appreciate the delicate figures which quite help me understand the studies. The analysis and conclusions are reasonable and the findings are inspiring, rehealing missed parts of NIC design and assessment. Overall I believe this paper can technically contribute to the community.

**Weaknesses:**

1. My major concern is the generalization of the conclusion. All the FR experiments are conducted on the hyperprior model (Ball'e et al., 2018), which is a somewhat outdated baseline. Many more powerful approaches have been proposed in the past years, involving more complex context models (Minnen et al., 2018, [40]), improved mixed quantization estimators ([e1, e2, e3]), and non-convolution network structures ([e2, e4, e5]). Shall we observe some different spectral properties when using those transformer-based models?
Also, the wavelet-based JPEG2000 is neither a SOTA handcraft codec nor a worldwide popular choice. Though conventional coders don't play key roles in this paper, I suggest the authors further discuss HEIF, AVIF or VVC(IF), and JPEG.
2. As discussed by the authors, the spectral preference of NIC may be strongly related to the MSE objective. This can be explained as it corresponds to a Gaussian likelihood, according to [Ball'e et al., 2018]. However, this loss function can be replaced to MS-SSIM, GAN and LPIPS etc. For instance, refer to perceptual optimized NIC approaches [e7]. Can this OOD drawback get suppressed when consider those objectives? More discussions are expected.
3. Though the paper provides a detailed analysis with proposed tools and benchmarks, It do not provide new approaches to solve the concerned OOD issues. Without more practical suggestions, I cannot strongly appreciate the findings for above-mentioned concern about generalization.

P.S. OOD in NIC is not a brand new topic. The authors may want to discuss more related works eg [e6].

[e1] Channel-wise autoregressive entropy models for learned image compression, Minnen et al.
[e2] Elic: Efficient learned image compression with unevenly grouped space-channel contextual adaptive coding, He et al.
[e3] Transformer-based transform coding, Zhu et al.
[e4] The Devil Is in the Details: Window-based Attention for Image Compression, Zou et al.
[e5] Learned image compression with mixed transformer-cnn architectures, Liu et al. *(concurrent work)*
[e6] On the Out-of-distribution Generalization of Probabilistic Image Modelling, https://openreview.net/pdf?id=q1yLPNF0UFV
[e7] High-fidelity generative image compression

**Questions:**

See above.

**Limitations:**

See above.

---

> ### Author Rebuttal · Authors · 2023-08-09
>
> Thank you for your thoughtful review of our submission. In addition to our inline responses to your comments below, please note that our global response highlights new experiments **E1** and **E2** and their insights **I1** - **I4** as well as clarifies the significance of our work in **C1** - **C3**. Namely, existing tools are incapable of highlighting inter- and intra-class similarities and differences for classical codecs and NICs. Our benchmark and introspection tools effectively uncover similarities and differences, in turn, providing a deeper understanding of their performance. Our tools will be available to researchers/practitioners to explore and improve NIC methods impacting future NIC research and development. After reflecting on our individual/global response, we hope that you will update your score and champion our paper for acceptance.
>
>
>
> > My major concern is the generalization of the conclusion. All the FR experiments are conducted on the hyperprior model (Ball'e et al., 2018), which is a somewhat outdated baseline. Many more powerful approaches have been proposed in the past years, involving more complex context models (Minnen et al., 2018, [40]), improved mixed quantization estimators ([e1, e2, e3]), and non-convolution network structures ([e2, e4, e5]). Shall we observe some different spectral properties when using those transformer-based models?
>
> Thank you for your suggestions. Per your request, we performed additional experiments with ELIC [e2]. Please refer to experiment **E2** and the subsequent insights **I1** - **I4** in the global response for takeaways from our experiments. We also started training LIC_TCM (from [e5]) on the CLIC dataset but it did not finish training in time to include in the rebuttal. We believe that these additional results strengthen our work and further highlight the usefulness of our benchmark and tools. We will also cite these papers in our future work section.
>
> > I suggest the authors further discuss HEIF, AVIF or VVC(IF), and JPEG.
>
> Per your request, we performed an evaluation of JPEG, noted as (**E1**) in our global response with visualizations in figures **P1**, **P2**, and **P5** in the rebuttal PDF. See insights **I1-I4** for takeaways pertaining to JPEG. A comparison to the VVC (equivalently VTM) codec was provided in Appendix J (noted in lines 141-143 of main paper). Importantly, our tools show that compression error from VTM is distributed more evenly across the frequency spectrum, unlike JPEG (added in **E1**) and JPEG2000, as illustrated in **Figures 16 & 17** (Appendix J) and **Figure P1**. We believe that these additional results strengthen our work and further highlight the usefulness of our benchmark and tools.
>
>
> > As discussed by the authors, the spectral preference of NIC may be strongly related to the MSE objective. This can be explained as it corresponds to a Gaussian likelihood, according to [Ball'e et al., 2018]. However, this loss function can be replaced to MS-SSIM, GAN and LPIPS etc. For instance, refer to perceptual optimized NIC approaches [e7]. Can this OOD drawback get suppressed when consider those objectives? More discussions are expected.
>
> Note that we provide a comparison to models optimized for MS-SSIM in Appendix G (as noted on lines 141-143 of the main paper). We agree that further study on the effects of alternative loss functions on robustness of NIC is of great interest. Also note that one cannot answer these excellent questions without our benchmark and tools  – highlighting the value of our work. We encourage the reviewer to recall the main contributions of our work in section (**GR2**) of our global response. We will consider listing robustness analysis of NICs trained with alternative loss functions in our future work section.
>
>
> > Though the paper provides a detailed analysis with proposed tools and benchmarks, It do not provide new approaches to solve the concerned OOD issues. Without more practical suggestions, I cannot strongly appreciate the findings for above-mentioned concern about generalization.
>
> Respectfully, it is our position that this view overlooks the meaningful contributions provided by our paper and grades them based on criteria we did not claim as contributions or efforts. Recall that, through our developed tools and experiments, we have evaluated the out-of-distribution robustness of neural/learned image compression providing a more sober assessment of its utility in real world scenarios. We agree with your sentiment that our findings pose new challenges for the NIC community; however, this is a benefit to the community as it should stimulate new and future research directions for NIC at the intersection of out-of-distribution generalization/robustness. Furthermore, our analysis and findings provide clear actionable guidelines. For starters, we have mentioned several approaches in the future work section of our paper. Additionally, tools from the OOD robustness community can be used (e.g., spectral data augmentation-based training, test-time adaptation of architectures for countering the shifts, etc.).
>
>
> > P.S. OOD in NIC is not a brand new topic. The authors may want to discuss more related works eg [e6].
>
> Respectfully, our work and the referenced work [e6] have significant differences. From our understanding, [e6] notices some similarities between *lossless* image compression and OOD *detection* and proposes a model which can work for both tasks. Our work differs in that we are studying *lossy* compression and the *robustness of lossy compression models to OOD images*, and we provide novel tools to make this study possible. The main similarity between our paper and [e6] are the keywords.

---

> > ### Author Response · Authors · 2023-08-20
> >
> > Dear Reviewer fKAX,
> >
> > We would like to thank you once again for your thoughtful initial review of our submission. We are fully aware of the dedication and time that your review entails and we deeply value the effort that you devoted to this.
> >
> > With only 1 day remaining in the discussion phase, we would be very grateful for any feedback you may have about our rebuttal, including both our global response with additional experiments and our individual response to your review. Your insights are of immense importance to us and we eagerly anticipate your updated evaluation. Should you find our responses informative and useful, we would be appreciate your acknowledgement. Furthermore, if you would like more clarification about a particular aspect of our response, please don’t hesitate to reach out. We are fully committed to providing additional responses during this crucial discussion phase.
> >
> > We sincerely thank you for your continued support and consideration. Your expertise is pivotal to the advancement of our research.
> >
> > Sincerely,
> >
> > Authors

---

### Official Review · Reviewer_h4BN · 2023-07-08

**Soundness:** 2 fair
**Presentation:** 2 fair
**Contribution:** 2 fair
**Rating:** 3
**Confidence:** 4

**Summary:**

This paper introduces a benchmark for evaluating the out-of-distribution (OOD) performance of neural image compression (NIC) models. The authors observe distinguished spectral biases of NIC models by comparing them with JPEG2000. Then the authors listed all the spectral biases along with a theoretical analysis from the perspective of principle component decomposition.

**Strengths:**

1. This paper proposes an interesting perspective for analyzing NIC models.

2. The observed spectral biases are potential to further improve NIC models.

3. The authors provide thorough analysis and enough details for reproducing the results.


**Weaknesses:**

I believe the spectral perspective is important to understand NIC models. However,  the theoretical analysis is of limited novelty and the analysis and conclusions provided in this paper are empirical and need to be further improved before publication. Moreover, I am afraid that classifications on several corruptions may conflict with the common sense in signal processing field. My concerns are listed as below.

1. The proposed method mainly compares with JPEG2000, which may be inappropriate. JPEG2000 is based on discrete wavelet transform. However, EBCOT and rate control of JPEG2000 could hinder an explicit understanding of the spectral characterization of JPEG2000. I suggest the authors to conduct the comparison experiments on JPEG (which is based on block-based DCT and simple quantization table) for clear comparison. Compared to JPEG2000, JPEG has been widely applied in practice.

2. The comparison presented in Figures 16 and 17 (i.e., comparison to VTM provided in supplementary material) is confusing. $\mathcal{D}$ for VTM seems to be zero but it is a lossy compression method and the reconstruction is still far from lossless (i.e., 37.42 dB).

3. I also have problem with the evaluation using power spectral density (SPD). I noticed that the authors classify Gaussian noise as a high-frequency corruption in Figure 1. However, it is well known that the Gaussian noise, i.e., white noise, has a constant SPD. In other words, the effect of Gaussian noise is expected to be constant rather than constrained on the high-frequency components. Moreover, the effect of JPEG compression is also expected to be on both median and high-frequency, as the default quantization map results in a larger quantization step on the high-frequency components. I am afraid that the classification of other corruptions may neither be sufficiently precise. I guess the gaps between above theoretical analysis and the empirical analysis in Figure 1 originate from the intensity difference on the SPD used to analyze the effect of introduced corruption. The natural images themselves are with an unbalanced SPD. Thus, the observation may be biased.

4. The theoretical analysis in Section 5 is not novel enough. The desired characteristics for transform coding are discussed in [R1]. The differences and similarity between DFT, DCT, and KLT are analyzed in [R2]. The provided analysis on linear transforms (autoencoder) may not be appropriate for nonlinear models [R3].

5. Considering that the major contribution of this paper lies on the empirical evaluations, it is important to make thorough validations on diverse datasets and NIC models. The adopted scale hyperprior is not sufficient supporting the claims in this paper.

[R1] V. K. Goyal, "Theoretical foundations of transform coding." IEEE Signal Processing Magazine, vol. 18, no. 5, pp. 9-21, 2001.

[R2] D. S. Taubman and M. W. Marcellin, JPEG2000: Image compression fundamentals, standards and practice. 2002.

[R3] J. Ballé, et al. "Nonlinear transform coding." IEEE Journal of Selected Topics in Signal Processing, vol. 15, no. 2, pp. 339-353, 2021.


**Questions:**

1. Could the authors offer a comparison with JPEG compressed images? I believe it would be more straightforward.

2. What is the reason for theoretical analysis (given in comment #3 in the section of Weakness) and the given classification (Figure 1) on Gaussian noise and JPEG compression artifacts?

3. What is the difference between the theories for nonlinear transforms (Section 5) and those for traditional linear transform (such as DWT, DCT, and KLT)?


**Limitations:**

Limitations are not discussed in the paper.

---

> ### Author Rebuttal · Authors · 2023-08-09
>
> Thank you for your thoughtful review of our submission. We believe that there is some misunderstanding in the goal and contribution of our work which we first clarify. Namely, existing tools are incapable of highlighting inter- and intra-class similarities and differences for classical codecs and NICs beyond RD curves. Our benchmark and introspection tools effectively uncover similarities and differences, in turn, providing a deeper understanding of their performance. Note that we found your feedback to be valuable to improve the impact and reception of our paper and, hence, we accommodated many of your suggestions. Specifically, our global response highlights additional requested experiments **E1**-**E2** and their insights **I1**-**I4** as well as clarifies the significance of our work in **C1**-**C3**. We believe these additional experiments enhance the comprehensiveness of the empirical evaluations, and further highlight the usefulness of our tools. After reviewing our individual and global responses, we hope that you will reflect on your review and consider updating your score.
>
> We now respond to your questions **Q** and weaknesses **W**.
>
> **Q1**
>
> Per your request, we performed an evaluation of JPEG, noted as **E1** in our global response with visualizations in figures **P1, P2, P5** in the rebuttal PDF. See insights **I1-I4** in (**GR1**) of the global response for JPEG takeaways.
>
> **Q2a** (Reason for theoretical analysis?)
>
> The theoretical results and takeaways (lines 323-340) provide support for the spectral bias of NIC beyond our empirical results with our spectral tools. We are not aware of theoretical results *explicitly purporting such a spectral bias of NICs* (as [R1]-[R3] do not). We welcome the reviewer to provide such references and we will revise our claim.
>
> **Q2b and W3** (Afraid classifications on several corruptions may conflict with signal processing. Reason for classification (Fig 1) on Gaussian noise and JPEG?)
>
> We believe this a misinterpretation of the classification provided in Table 1, which we clarify here. Firstly, we are using characterizations of corruptions into low, medium, and high from [Li19] (Apdx Table 1). This characterization has been adopted by the neural network robustness community and has shown its practical utility by showing that these corruptions classes behave differently. This is further supported by our empirical results. Next, we clarify the classification.
>
> Categorizing a corruption as “high” is more akin to saying it “contains substantial high-frequency content” (but of course doesn’t exclusively consist of high-frequency content). This can be easily resolved by elaborating on the classification into low, medium, and high frequency groups which we will clarify in our paper and reference where it originated. We would even be willing to change the name of these categories if the reviewer suggests so, as doing so does not affect our theoretical and empirical claims.
>
> **Q3 and W4a** (Difference/similarity between DFT, DCT, KLT analyzed in [R2])
>
> At the risk of oversimplifying, DFT, DCT and KLT were hand crafted by experts based on understanding of image statistics. Our linear autoencoder example suggests that NIC models may *learn* a somewhat similar compression technique **directly from training data**. On the other hand, the more general analysis of possibly nonlinear models in Appendix N suggests among other things that (not surprisingly) MSE-optimized autoencoders suffer high error in regions of low training data density. For more fine-grained details we refer to that section.
>
> **W1**
>
> We note that our empirical results are already highlighting the differences and value (so not hindering an explicit understanding of the spectral characterization of JPEG2000). While EBCOT and rate control of JPEG2000 might hinder a theoretical understanding of its spectral properties, we compute PSDs empirically – could you elaborate on how they could obstruct an empirical spectral characterization of JPEG2000 and make our claims invalid?
>
> **W2**
>
> We agree that VTM is lossy compression and Figs 16/17 reflect this. To clarify, note that the PSDs in Fig 16 utilize the same heatmap range for a fair comparison across methods while the PSDs in Fig 17 have the default heatmap range, [0,max PSD value]. Importantly, our tools show that distortion from VTM is distributed more evenly across the frequency spectrum, unlike JPEG (added in **E1**) and JPEG2000, as illustrated in Figs 16 & 17 (Appendix J) and **Fig P1**.
>
> **W3**
>
> You are of course correct that Gaussian noise is evenly distributed in frequency space (as supported by Fig 20 in Appendix K, and the new **Fig P4**). Similarly, your statement about JPEG is correct and corroborated by our JPEG PSD in the rebuttal PDF, **Fig P4**. This confusion has arisen from differing interpretations of the corruption categories from the robustness community [Li19]. Further, this does not affect any of our empirical claims in the paper.
>
> **W4b** (Analysis on linear…)
>
> We provide a separate theoretical analysis that applies to nonlinear models in Appendix N (as noted on line 303 of paper).
>
> **W5**
>
> Respectfully, it is our position that this view overlooks the major meaningful contribution provided by our paper: Our benchmark and introspection tools – please see section (**GR2**) of the global response. However, we acknowledge that the utility of our benchmark and tools can be better highlighted through the inclusion of additional NIC models. To this end, we prioritized training and evaluating an additional model on CLIC, ELIC [He22], during the rebuttal period. Please see section (**GR1**) of the global response for details and insights from these experiments **I1**-**I4**. Of note, ELIC demonstrates similar spectral bias to the Balle NIC model included in our original paper. We believe these additional experiments enhance the quality of the empirical evaluations and highlight the usefulness of our tools.

---

> > ### Author Response · Authors · 2023-08-20
> >
> > Dear Reviewer h4bn,
> >
> > We would like to thank you once again for your thoughtful initial review of our submission. We are fully aware of the dedication and time that your review entails and we deeply value the effort that you devoted to this.
> >
> > With only 1 day remaining in the discussion phase, we would be very grateful for any feedback you may have about our rebuttal, including both our global response with additional experiments and our individual response to your review. Your insights are of immense importance to us and we eagerly anticipate your updated evaluation. Should you find our responses informative and useful, we would be appreciate your acknowledgement. Furthermore, if you would like more clarification about a particular aspect of our response, please don’t hesitate to reach out. We are fully committed to providing additional responses during this crucial discussion phase.
> >
> > We sincerely thank you for your continued support and consideration. Your expertise is pivotal to the advancement of our research.
> >
> > Sincerely,
> >
> > Authors

---

> > ### Comment · Reviewer_h4BN · 2023-08-21
> >
> > Thanks for the authors' hard work during the rebuttal. The detailed response and additional experiments are impressive. However, the main findings in the previous version seems to become less convincing as the spectral difference between the two NIC models (i.e., FR NIC and ELIC in Figure P1) could also be distinguishing. I also notice that the authors modified their statement about the main findings. The original paper addresses the difference between NIC and traditional codecs (Lines 62-77), while the response (I1-I4 in Author Rebuttal by Authors) addresses the differences across diverse codecs. Therefore, I maintain my view that this paper needs to be further revised before publication and I am to keep my score.

---

> > > ### Author Response · Authors · 2023-08-21
> > >
> > > ​​Thank you for taking the time to review our results. We hope and assume that our response clarified your initial concerns. Respectfully, we are confused by your statement that “the main findings in the previous version seems to become less convincing” because one can utilize our tools to analyze and compare **any** codec/NIC. In fact, we already compared two NIC methods in the initial version. Thus, our message and claims are still the same as before. We’d like to emphasize again that our contributions are as listed in **C1-C3** of the joint response. Specifically, our novel tools and benchmarks reveal previously unknown similarities and differences between all methods of image compression. We are not providing a comprehensive evaluation over all image compression methods, but do include comparison between several methods as examples of how these tools can be used and how the spectral bias of NIC can largely alter performance on OOD tasks. We perceive highlighting such fine-grained similarities and differences as our strength and not weakness. Additionally, we did not intentionally alter our statement about the main findings. These differences come from the fact that we went from comparing two specific methods (JPEG2000 and Balle) to comparing a more general set of codecs.

---

### Official Review · Reviewer_e3od · 2023-07-09

**Soundness:** 4 excellent
**Presentation:** 4 excellent
**Contribution:** 2 fair
**Rating:** 5
**Confidence:** 3

**Summary:**

This paper analyzes neural image compression models relative to classical (hand-engineered) methods using a new dataset and new tools based on spectral analysis. Specifically the paper presents (from Section 1, "Main Contributions"):
(1) a new dataset covering different kinds of distortions used to compare methods
(2) an inspection tool for distortions that shows where, in the frequency domain, different distortions primarily introduce error
(3) an inspection tool based on spectral analysis that provides a visualization showing which frequency ranges are most distorted by different compression methods
(4) some theoretical results about neural compression, albeit limited to the linear autoencoder case

The main findings are also summarized in the intro:
(1) Neural image compression (NIC) produces different spectral artifacts than classical codecs (JPEG 2000 was used as the classical codec and a scale-hyperprior model was used for NIC)
(2) NIC and classical codecs prioritize different frequencies as the bit rate changes
(3) NIC and classical codecs fail to generalize in similar ways but succeed in different ways
(4) Due to the differences, it can be hard to choose the best codec for a particular application without knowing what kind of spectral distortions will be present at inference time (i.e., out of distribution)

**Strengths:**

Although compression systems, neural and hand-engineered, are frequently assessed in terms of visual quality, this paper is the first I've seen that tries to understand how distortions differ between neural and standard methods. The authors develop a new dataset built by applying 15 different kinds of distortions to images from standard evaluation sets (kodak and clic). They then visualize the power spectral density (PSD) of the distorted images (to understand which frequency bands are affected by each type of distortion) and they visualize the distortions caused by lossy compression for a neural method and for JPEG 2000.

This analysis leads to interesting observations about how the two codecs behave differently, e.g. neural image codecs (NICs) distort high frequencies more than low frequencies (Section 4.1), and they are more sensitive to high frequency noise/distortions. I haven't previously this kind of analysis or conclusion before.

Similarly, the authors analyze how classification accuracy degrades when the two types of codecs are used on images with different kinds of distortion (Fig. 5). These findings are particularly interesting because neither codec is preferable for all distortions and bit rates, e.g., JPEG2000 takes a large hit for "high" distortions at high bit rates but takes a smaller hit than NICs for "low" distortions at moderate bit rates.

**Weaknesses:**

There are two main weaknesses with the paper. The first is whether the results from studying these two codecs generalize to other codecs in the respective classes. For example, will VCC (a modern codec still be finalized) behave similarly to JPEG2000? Will SOTA neural methods behave similarly to [9]. The latter question is further complicated because there are multiple ways the NICs differ, e.g. (1) more sophisticated models that build on [9] (like adding autoregressive priors, architectures built using transformers instead of CNNs, etc.), (2) using "implicit neural representations" instead of the "nonlinear transform coding" approach of [9], and (3) optimizing networks with an adversarial loss that targets "realism" (minimizing a divergence between distributions, not just a measure of distortion between the original and reconstructed pixel values). It's hard to say if the findings in this paper will be the same for a model that is different according to one of (or all of) these dimensions.

Second, I'm not sure how important these findings are to the NeurIPS community. Neural image compression is certainly an appropriate topic for NeurIPS, but papers typically focus on new models or new applications. So the target audience may be fairly small, and the observations don't directly imply an architecture change (or loss / optimization change) to improve NICs.

Since I've worked in this subfield, I find the results of this paper fascinating. And at a conference focused on compression or image processing (something like ICIP or DCC), I'd rate this paper as a 7+. I'm open to feedback from the AC or other reviewers that I'm off on my assessment of the fit at NeurIPS.

**Questions:**

No questions beyond the implied ones in the "weaknesses" section.

**Limitations:**

adequately covered

---

> ### Author Rebuttal · Authors · 2023-08-09
>
> Thank you for your thoughtful review of our submission. We are grateful to hear an expert in this subfield found our work to be fascinating while being mainly concerned about the fit of our work at NeurIPS. We appreciate you stating clear reasoning behind your rating as it gives us an opportunity to clarify any confusion. In addition to our inline responses to your comments below, please note that our global response highlights additional experiments **E1** and **E2** and their insights **I1** - **I4** as well as clarifies the significance of our work in **C1** - **C3**. Namely, existing tools are incapable of highlighting inter- and intra-class similarities and differences for classical codecs and NICs. Our benchmark and introspection tools effectively uncover similarities and differences, in turn, providing a deeper understanding of their performance. Our tools will be available to researchers/practitioners to explore and improve NIC methods impacting future NIC research and development. We hope that you will update your score to reflect your enthusiasm for our work and champion our paper for acceptance.
>
> > Two main weaknesses … similarly to JPEG2000?
>
> Respectfully, this question highlights the utility of our introspection tools and is not a weakness of our paper. Specifically, our evaluation tools highlight both inter- and intra-class similarities and dissimilarities that might not be possible to uncover using existing evaluation datasets and scalar metrics. Importantly, our tools show that distortion from VTM (equivalently VVC; instead of VCC which we took to be a typo) is distributed more evenly across the frequency spectrum, unlike JPEG (added in **E1**) and JPEG2000, as illustrated in **Figures 16 & 17** (Appendix J) and **Figure P1**. Namely, our tools can be utilized to analyze and compare any combination of codecs/NICs, either presently available or developed in the future and can be used to guide the design of future compression schemes.
>
> > Will SOTA neural methods behave similarly to [9]? The latter … these dimensions.
>
> We agree that these are interesting questions and again believe these highlight the utility of our work. Our tools allow researchers to identify more nuanced differences between NIC variants, even if they have similar performance in terms of RD curves, which are the main existing metrics.
>
> More specifically, we do include some additional comparisons of NIC variants. On lines 141-143 of the main paper we note that comparisons with NIC optimized for MS-SSIM are available in Appendix G. Additionally, during the rebuttal period we performed empirical analysis for an additional SOTA NIC method, ELIC [He22], trained on CLIC.
>
> > I'm not sure how important findings are to NeurIPS community and observations don't directly imply NIC changes
>
> We appreciate your perspective and agree that significant NIC contributions in the past have advanced NIC architectures/loss/etc. First, note that we have clarified the significance of our work in section (**GR2**) of the global response. Our work not only highlights future algorithmic directions for NIC but our benchmark and tools provide the means to perform meaningful evaluation that can guide new research.
>
> Additionally, our choice of NeurIPS as a venue was based on the following bullet points from the official call for papers:
>
> * Applications (e.g., vision, language, speech, audio)
> * Evaluation (e.g., methodology, meta studies, replicability, validity)
> * Machine learning for sciences (e.g. climate, health, life sciences, physics, social sciences)
> * Social/economic aspects of ML (e.g., fairness, interpretability, human-AI interaction, privacy, safety, strategic behavior).
>
> We submitted under the primary area of "Evaluation" as our paper a) points out major discrepancies between models with the same PSNR/bpp and b) proposes new metrics/datasets to give researchers a more comprehensive understanding of these differences.
>
> > Target audience may be fairly small
>
> Our work is relevant to both NIC researchers and the broader ML safety and robustness research community. When combined, these two research groups are a significant audience. Further, it has the potential to create a new audience at the intersection of NIC and out-of-distribution generalization/robustness.
>
> > Authors present some theoretical results about neural compression limited to linear autoencoders.
>
> A result applying to nonlinear models is provided in Appendix N (noted on line 303 of main body). We included the linear autoencoder case in the main body since it is less technically involved, and the qualitative takeaways are similar to those in Appendix N.

---

> > ### Author Response · Authors · 2023-08-20
> >
> > Dear Reviewer e3od,
> >
> > We would like to thank you once again for your thoughtful initial review of our submission. We are fully aware of the dedication and time that your review entails and we deeply value the effort that you devoted to this.
> >
> > With only 1 day remaining in the discussion phase, we would be very grateful for any feedback you may have about our rebuttal, including both our global response with additional experiments and our individual response to your review. Your insights are of immense importance to us and we eagerly anticipate your updated evaluation. Should you find our responses informative and useful, we would be appreciate your acknowledgement. Furthermore, if you would like more clarification about a particular aspect of our response, please don’t hesitate to reach out. We are fully committed to providing additional responses during this crucial discussion phase.
> >
> > We sincerely thank you for your continued support and consideration. Your expertise is pivotal to the advancement of our research.
> >
> >
> > Sincerely,
> >
> > Authors

---

> > ### Comment · Reviewer_e3od · 2023-08-22
> >
> > > this question highlights the utility of our introspection tools and is not a weakness of our paper.
> >
> > Yes, I agree with this point. I think my framing was that if we wanted to draw conclusions between NIC and standard methods as two classes of methods, we may not see clear distinctions since there's so much variation within each category. The authors seem to say the same thing in their response (GR2.C3):
> >
> > [Our work] demonstrates the utility of our datasets and tools by helping uncover both inter- and intra-class (classes = {classical codecs, NIC models}) similarities and differences among existing methods
> >
> > So the question is if we analyze more methods from the two groups, do the initial inter-class similarities and differences hold or break down? The authors added evaluations of JPEG and ELIC so coverage is better. Honestly, I'm not sure from the new results if the initial similarities/difference hold.
> >
> > Regardless, we can see value in the tools and datasets regardless of clear inter/intra-class similarities/differences. I think this is the authors' point in the rebuttal? From this perspective, their tools will provide insight into any method so it doesn't matter if VVC and JPEG2000 behave similarly.
> >
> > > do include some additional comparisons of NIC variants
> >
> > The authors added ELIC, which strengthens the paper. I mentioned INRs in my initial review but commented above (in response to another review) that an evaluation of an INR-based method should not hurt the papers' acceptance chance. I wanted to call that out here to be consistent.
> >
> > > choice of NeurIPS as a venue
> >
> > The authors make a fair point about NeurIPS and cite the "Evaluation" section from the official call.
> >
> >
> > Based on the author's new results and rebuttal (and extensive appendix), I'll revise my rating up.

---

### Official Review · Reviewer_5yiv · 2023-07-13

**Soundness:** 3 good
**Presentation:** 4 excellent
**Contribution:** 3 good
**Rating:** 6
**Confidence:** 4

**Summary:**

This paper addressed the problem of image compression. Neural-based methods have started outperforming traditional codec methods in image compression, but there are no comprehensive datasets and informative tools to evaluate neural-based methods in real-world settings. This paper proposed a benchmark to study this that allows for evaluating out-of-distribution performance. The proposed benchmark provides some findings that are crucial to bridge the real-world adoption of neural-based image compression techniques.

**Strengths:**

1. Comprehensive datasets for image compression.
2. Understanding of out-of-distribution performance.
3. Extensive evaluations.
4. Theoretical analyses.

**Weaknesses:**

NeRF-based methods can also be seen as a way to do compression (compressing images into. network weights), The authors should also discuss this and benchmark them on the dataset.

**Questions:**

Overall, this is a strong paper. I only have some minor comments. See above.

**Limitations:**

The paper should address this.

---

> ### Author Rebuttal · Authors · 2023-08-09
>
> Thank you for your thoughtful review of our submission. In addition to our inline response to your comment below, please note that our global response highlights additional experiments **E1** and **E2** and their insights **I1** - **I4** as well as clarifies the significance of our work in **C1** - **C3**. Namely, existing tools are incapable of highlighting inter- and intra-class similarities and differences for classical codecs and NIC models. Our benchmark and inspection tools effectively uncover similarities and differences, in turn, providing a deeper understanding of their performance. Our tools will be available to researchers/practitioners to explore and improve NIC methods impacting future NIC research and development. After reflecting on our individual and global responses, we hope that you will update your score and champion our paper for acceptance.
>
> > “NeRF-based methods can also be seen as a way to do compression (compressing images into network weights), The authors should also discuss this and benchmark them on the dataset.”
>
> Thank you for this suggestion. We agree that testing a wider variety of compression methods would strengthen the empirical side of our paper. During the rebuttal period, we prioritized adding an additional “mainstream” NIC algorithm (ELIC) and standard codec (ordinary JPEG) (experiments **E1** and **E2** in our global response). While we think comparison with NeRF compression would be very interesting, we hope you will understand that adding it would require significant expansion of our research code-base and an entirely different theoretical analysis, and as such would be better suited for a follow-up project. We have added it as a future direction in the conclusion. Finally, we would like to emphasize again that our datasets and spectral tools enable the community to ask and answer these questions, which clearly demonstrates the value of our work.
>
>
> > Limitations: The paper should address this.
>
> We will add an explicit limitations section to address this.

---

> > ### Author Response · Authors · 2023-08-20
> >
> > Dear Reviewer 5yiv,
> >
> > We would like to thank you once again for your thoughtful initial review of our submission. We are fully aware of the dedication and time that your review entails and we deeply value the effort that you devoted to this.
> >
> > With only 1 day remaining in the discussion phase, we would be very grateful for any feedback you may have about our rebuttal, including both our global response with additional experiments and our individual response to your review. Your insights are of immense importance to us and we eagerly anticipate your updated evaluation. Should you find our responses informative and useful, we would be appreciate your acknowledgement. Furthermore, if you would like more clarification about a particular aspect of our response, please don’t hesitate to reach out. We are fully committed to providing additional responses during this crucial discussion phase.
> >
> > We sincerely thank you for your continued support and consideration. Your expertise is pivotal to the advancement of our research.
> >
> > Sincerely,
> >
> > Authors

---

> > ### Comment · Reviewer_e3od · 2023-08-22
> > **Missing comparison to "implicit" methods doesn't hurt the paper**
> >
> > Nerf-like approaches, often called implicit neural representations (INRs), are very interesting. All of the literature I've seen (as of a few months ago, at least), show that this approach trails methods based on autoencoders on typical metrics. So comparing against a different approach to neural compression is interesting, but, in my opinion, isn't vital and so the absence of this comparison shouldn't reduce the paper's chance of acceptance.

---

### Author Rebuttal · Authors · 2023-08-09

We thank the reviewers for their valuable feedback. We are grateful that reviewers found our contributions for benchmarking neural image compression models to be “comprehensive” (5yiv), “interesting” (e3od, h4BN), “important” (h4BN), “fascinating” (e3od), “crucial to bridge the real-world adoption of neural-based image compression techniques” (5yiv), and providing “potential to further improve NIC models” (h4BN). Several reviewers found our empirical evaluation and subsequent findings to be “extensive” (5yiv), “inspiring” (fKAX), and providing an “understanding of out-of-distribution performance” (5yiv). Furthermore, reviewers indicated our contributions to NIC were “the first [they have] seen that [try] to understand how distortions differ between neural and standard methods” (e3od) and that it serves to “[reveal] missed parts of NIC design and assessment” (fKAX).

(**GR1**) While some reviewers found our empirical (5yiv, fKAX) and theoretical (5yiv) analysis a strength, others requested additional results. Some of these requested results, such as comparison to an additional codec (VVC, equivalently VTM), NIC optimized for MS-SSIM, and theoretical results for nonlinear autoencoders were provided in the appendix (Appendix J, Appendix G, Appendix N) and noted in the main paper (lines 141-143 and line 303). Furthermore, during the rebuttal period we obtained new results for the remaining experiments which are included in the rebuttal PDF:

* **E1** Evaluation of JPEG codec (fKAX, h4BN) in figures **P1**, **P2**, and **P5**.
* **E2** Evaluation using additional NIC architecture (fKAX, h4BN, e3od) [ELIC](https://github.com/VincentChandelier/ELiC-ReImplemetation) [He22] in figures **P1**, **P2**, **P3**, and **P5**. Note that we trained ELIC on the CLIC dataset during the rebuttal period (as opposed to using the publicly-available ImageNet-trained weights).

Notably, **E1** and **E2** align with our original insights offered by our benchmark and tools and serve to strengthen our claims (h4BN, fKAX), namely that:

* **I1** Compression methods yielding the same PSNR (or bpp) can produce very different spectral artifacts (**Figure P1**)
* **I2** As the compression rate increases, different codecs prioritize different parts of the spectrum (**Figures P1 and P3**)
* **I3** Image compression models generalize to low- and mid-frequency shifts better than high- frequency shifts (**Figures P2 and P5**)
* **I4** NIC models are better at denoising high-frequency corruptions than classic codecs (**Figures P2 and P5**)



(**GR2**) In conclusion, we would also like to emphasize that our work:

* **C1** Highlights limitations of existing evaluation benchmarks and tools for image compression
* **C2** Provides new benchmarks and tools to gain a deeper understanding of compression performance
* **C3** Demonstrates the utility of our datasets and tools by helping uncover both inter- and intra-class (classes = {classical codecs, NIC models}) similarities and differences among existing methods

Comprehensive evaluation schemes are critical to make sustainable progress in any research area, and it seems to be generally acknowledged that test sets measuring robustness to distribution shift have been crucial to the image classification and language modeling research areas (e.g., ImageNet-{C,R,A,v2}, HANS). Our work brings these methods to bear on image compression while providing new insights on the spectral biases of existing codecs and NICs. We are hopeful that reviewers will find our responses clarifying and that, upon reflection of our work’s meaningful contributions to ongoing and future NIC research directions, they will champion our work for acceptance.



**References**
* [He22] He et. al., *ELIC: Efficient learned image compression with unevenly grouped space-channel contextual adaptive coding*, CVPR 2022.
* [Li19] Li et. al., *Robust deep learning object recognition models rely on low frequency information in natural images*, PLOS Computational Biology 19(3).
* [Vardi23] Gal Vardi, *On the Implicit Bias in Deep-Learning Algorithms*, Communications of the ACM, June 2023.

---

### Decision · Program_Chairs · 2023-09-21

**Decision:**

Accept (poster)

**Comment:**

Initial reviews were mixed with a wide spread in rating, author rebuttal helped address majority of the concerns. Although, one reviewer remains unconvinced, others have updated their ratings and are leaning positive to various extents. After reading the author rebuttal and reviewer discussion, I think that majority of the show stopping concerns were addressed and therefore recommend acceptance. I suggest that authors incorporate all the feedback to further improve their manuscript.